# Latent Veracity Inference for Identifying Errors in Stepwise Reasoning

**Minsu Kim**[1,2]* **Jean-Pierre Falet**[1,3,4]* **Oliver E. Richardson**[1,3] **Xiaoyin Chen**[1,3]
**Moksh Jain**[1,3] **Sungjin Ahn**[2] **Sungsoo Ahn**[2] **Yoshua Bengio**[1,3,4]
[1]Mila – Québec AI Institute [2]KAIST [3]Université de Montréal [4]LawZero
{minsu.kim, jean-pierre.falet}@mila.quebec

## Abstract

Chain-of-Thought (CoT) reasoning has advanced the capabilities and transparency of language models (LMs); however, reasoning chains can contain inaccurate statements that reduce performance and trustworthiness. To address this, we propose to augment each reasoning step in a CoT with a latent veracity (or correctness) variable. To efficiently explore this expanded space, we introduce *Veracity Search* (VS), a discrete search algorithm over veracity assignments. It performs otherwise intractable inference in the posterior distribution over latent veracity values by leveraging the LM's joint likelihood over veracity and the final answer as a proxy reward. This efficient inference-time verification method facilitates supervised fine-tuning of an *Amortized Veracity Inference* (AVI) machine by providing pseudo-labels for veracity. AVI generalizes VS, enabling accurate zero-shot veracity inference in novel contexts. Empirical results demonstrate that VS reliably identifies errors in logical (PRONTOQA), mathematical (GSM8K), and commonsense (COMMONSENSEQA) reasoning benchmarks, with AVI achieving comparable zero-shot accuracy. Finally, we demonstrate the utility of latent veracity inference for providing feedback during self-correction and self-improvement.

## 1 Introduction

The inference-time compute paradigm—the practice of allowing language models (LMs) to generate a chain-of-thought (CoT) before producing an answer—has lead to a improvements in reasoning performance across a variety of domains (Nye et al., 2021; Kojima et al., 2022; Wei et al., 2022). The CoTs themselves also promise a degree of interpretability, giving operators a tool to detect problematic behavior (Perez et al., 2023). This promise, however, is undermined by the fact that LMs often generate flawed reasoning steps (Ji et al., 2023; Zhang et al., 2024b; Bang et al., 2023). Flawed reasoning impairs interpretability and may propagate to the model's final output (Cobbe et al., 2021; Zelikman et al., 2022; Wang et al., 2023; Yao et al., 2023; Turpin et al., 2023; Lightman et al., 2024), making error detection (and correction) an important challenge for improving LM trustworthiness.

Several methods have been proposed to improve the correctness, or *veracity*, of a CoT. Training on labeled reasoning steps is one solution (Camburu et al., 2018; Rajani et al., 2019; Lightman et al., 2024), but it is impeded by the paucity of comprehensive annotated datasets due to high labeling cost. Fact verification through evidence retrieval from external corpora represents a compromise in terms of labeling requirements (Chern et al., 2023; Min et al., 2023; Jacovi et al., 2024), but faces other obstacles including retrieval complexity and evidence coverage gaps.

We propose a new method to automatically identify stepwise errors in a CoT without requiring supervision for each reasoning step. Our key idea is to formulate the problem of identifying errors as the problem of doing posterior inference in a latent-variable model (LVM) where each reasoning step is augmented with a latent veracity variable indicating its correctness. This label can be binary (True or False) for many applications, but our framework is also compatible with categorical variables that can take on more than two values. The CoT itself, along with the final output of the reasoning process (the answer to a query), are treated as observations that serve as the main signal for inferring accurate latent-veracity assignments. In more detail, our contributions are as follows:

---

*Equal contribution.

**Section 2.1:** We cast stepwise error identification as a latent-variable modeling problem.

**Section 2.2:** We introduce a discrete search algorithm, Veracity Search (VS), which leverages the LM's joint likelihood over stepwise veracity and the final answer as a proxy reward for approximately sampling from the the target distribution over latent veracity assignments, highlighting differences with standard methods that use in-context learning to turn LMs into verifiers.

**Section 2.3:** We propose Amortized Veracity Inference (AVI) to train an LM to predict a distribution over veracity assignments that does not depend on the true answer, using supervised fine-tuning on pseudo-labels obtained from VS. As a result, AVI enables zero-shot latent-veracity inference in downstream reasoning tasks where the final answer is unknown.

**Section 4:** We validate our approach on the logical reasoning benchmark PRONTOQA (Saparov & He, 2022), the mathematical reasoning task GSM8K (Cobbe et al., 2021), and commonsense reasoning COMMONSENSEQA (Talmor et al., 2018) using several open-source LMs (Qwen (Bai et al., 2023; Yang et al., 2024a;b), Llama (Touvron et al., 2023a;b; Grattafiori et al., 2024)). Our method yields consistent improvements in verification accuracy over in-context learning baselines, and scales to longer and more complex reasoning chains. We demonstrate the utility of error-identification in downstream tasks by using veracity assignments as feedback for self-correction and self-improvement (Pan et al., 2024).

## 2 METHOD

Let $P_{\mathrm{LM}}$ denote an LM's probability distribution over the set of possible sequences of tokens. Modern "thinking" LMs process an input prompt $x$ by first generating a CoT $z$ and subsequently producing the final answer $y$. Marginalizing out the CoT, the model's distribution of outputs given the input $x$ is $\mathbb{P}(y \mid x) = \sum_z P_{\mathrm{LM}}(y\, z \mid x) = \mathbb{E}_{z \sim P_{\mathrm{LM}}(z \mid x)} \left[ P_{\mathrm{LM}}(y \mid x\, z) \right]$. (Note that we use a different symbol, $\mathbb{P}$, to distinguish this probabilistic model from the result $P_{\mathrm{LM}}(y \mid x)$ of directly querying the LM.) Some approaches (e.g., Zelikman et al. (2022); Hu et al. (2024)) replace $P_{\mathrm{LM}}(z \mid x)$ with a learned distribution $Q(z \mid x)$ that puts more weight on "correct" reasoning chains by training $Q$ to approximate the distribution $\mathbb{P}(z \mid x, y^*)$, where $y^*$ is the true answer to the query. Underpinning these approaches is the assumption that $\mathbb{P}(y^* \mid x)$ is increased as a result of marginalizing with respect to such a $Q$. Typical strategies for training $Q$ include (i) supervised fine-tuning on labeled examples of $(x, z, y)$ (Gulcehre et al., 2023), and (ii) reinforcement learning (e.g., REINFORCE (Zelikman et al., 2022)), amortized inference (e.g., GFlowNets (Hu et al., 2024)), or test-time inference (e.g., Sequential Monte Carlo (Zhao et al., 2024)) using $P_{\mathrm{LM}}(z\, y^* \mid x)$ as a reward, to approximately sample from the intractable posterior $\mathbb{P}(z \mid x, y^*)$. An analogue of this distribution $Q$ lies at the heart of our proposal, but to explain it, we must first formally introduce an additional dimension: veracity.

### 2.1 A LATENT-VARIABLE MODEL ($\mathbb{P}$) AUGMENTED WITH VERACITY ($V_z$)

One should not expect every reasoning step in a CoT to be correct. Yet, by viewing $z$ as a logical statement and conditioning on it, standard practice implicitly identifies $z$ with the proposition that "$z$ is correct". From this starting point, it is clear why so much work has gone into correcting reasoning chains: so doing would validate the unstated assumption. Our approach is different: we take the *identity* of the CoT $z$ as given and introduce a new binary variable $V_z$ intended to capture the *veracity* of $z$. Since the standard assumption that $V_z$ is always 1 does not always hold in practice, we claim that identifying incorrect explanations $z$ is crucial.

As commonly done in stepwise CoT evaluation (Golovneva et al., 2023; Lightman et al., 2024; Manakul et al., 2023), we parse the CoT $z$ into a sequence of primitive statements $z = (z_1, z_2, \ldots, z_N)$; as a result, boolean veracity $V_z$ is a binary random vector taking values in $\{0, 1\}^N$. We imagine that the distribution over $V_z$ and $Y$ given $x$ and $z$ is governed by the behavior of an auto-regressive LM, which gives us a (conditioned) LVM $\mathbb{P}$ over the variables $V_z$ and $Y$:

$$\mathbb{P}(V_z = v, Y = y \mid x, z) \ := \ P_{\mathrm{LM}}(v\, y \mid x\, z) \ = \ P_{\mathrm{LM}}(v \mid x\, z)\, P_{\mathrm{LM}}(y \mid x\, z\, v). \tag{1}$$

The veracity variables $V_z$, however, are not observed, so calculating $\mathbb{P}(V_z \mid x, z, y)$ requires summing over the $2^N$ possible values of $V_z$ to calculate the denominator $\mathbb{P}(Y \mid x, z)$:

$$\mathbb{P}(V_z = v \mid Y = y, x, z) = \frac{\mathbb{P}(V_z = v, Y = y \mid x\, z)}{\mathbb{P}(Y = y \mid x\, z)} = \frac{P_{\mathrm{LM}}(v \mid x\, z) P_{\mathrm{LM}}(y \mid x\, z\, v)}{\sum_{v' \in \{0,1\}^N} P_{\mathrm{LM}}(v' \mid x\, z) P_{\mathrm{LM}}(y \mid x\, z\, v')}, \tag{2}$$

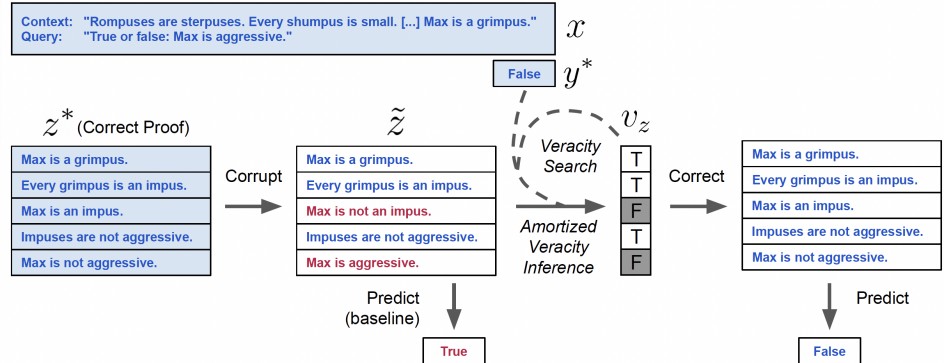

Figure 1: **Overview of our latent veracity inference method applied to PRONTOQA.** Given an input $x$, the *Veracity Search* (VS) takes an erroneous CoT $\tilde{z}$ and searches for a veracity vector $v_z$ with high joint likelihood $P_{\mathrm{LM}}(v_z \ y^* \mid x \ z)$, where $y^*$ is the correct answer. Veracity vectors can then be used as pseudo-labels to fine tune an LM via *Amortized Veracity Inference* (AVI) for zero-shot veracity inference, eliminating the dependencies on $y^*$ and on the test-time search algorithm (dotted lines). A LM can use veracity assignments for correcting flawed reasoning steps.

From another angle, one might view the problem as about the fixed-order auto-regressive nature of the LM, which makes joint probabilities under $P_{\mathrm{LM}}$ sensitive to the order of the sequence. If we assume the joint factorization in Eq. 2, corresponding to the order one would expect to sample veracity in if it is to be useful for predicting $Y$, then the problem of doing inference in a conditional LVM $\mathbb{P}$ in which $Y$ is observed but $V_z$ is not amounts to infilling $V_z$ in the sequence $X \to Z \to (V_z) \to Y$. For this reason, $\mathbb{P}(V_z{=}v \mid Y{=}y, x, z)$ is likely to differ from $P_{\mathrm{LM}}(v \mid x \ z \ y)$; the latter corresponds to an alternative generative model $X \to Z \to Y \to V_Z$, and may not be a good approximation the posterior $\mathbb{P}(V_z \mid x, z, y)$ of interest in our LVM. Nevertheless, $P_{\mathrm{LM}}(V_z{=}v \mid x \ z \ y)$ corresponds to an obvious in-context learning baseline where the LM is prompted to predict the veracity of a reasoning chain $z$. We evaluate against such baselines in Section 4.

Tying this back to the related work referenced at the top of §2, we therefore seek a variational posterior distribution $Q(V_z \mid x, z)$ over veracity values that puts more weight on "correct" veracity assignments by training it to approximate the intractable posterior $\mathbb{P}(V_z \mid x, z, y^*)$. The difficulty of doing so motivates us to design an efficient search-based approach that performs iterative refinement of $V_z$ (VS; §2.2), which will ultimately be the stepping stone to get such a $Q$ using AVI (§2.3). An overview of our approach in the context of a logical reasoning task is summarized in Fig. 1.

## 2.2 VERACITY SEARCH (VS)

Assume that the query $x$, CoT $z$ (possibly with errors), and—for now—the correct answer $y^*$ are given. Define the *proxy reward* for a bit vector $v \in \{0, 1\}^N$ as

$$R(v) \ := \ \mathbb{P}(V_z{=}v, Y{=}y^* \mid x, z) = P_{\mathrm{LM}}(v \ y^* \mid x \ z) \propto \mathbb{P}(V_z{=}v \mid Y{=}y^*, x, z), \tag{3}$$

where the proportionality relation is obtained via the application of Bayes rule in Eq. 2. VS seeks high-reward assignments $v_z \in \{0, 1\}^N$, which corresponds to sampling from $\mathbb{P}(V_z \mid x, z, y^*)$, the latent variable model's posterior over veracity assignments. In comparison to methods that produce better reasoning chains using a reward signal coming only from the target answer $Y$ (Cobbe et al., 2021; Zelikman et al., 2022), or that treats the CoT as a latent variable $Z$ that entangles identity and veracity (Hu et al., 2024; Phan et al., 2023), our proxy reward is taken from a latent variable model relating both veracity and final answers, and enables fixing the identity of the CoT $Z = z$ and focusing on the sub-problem of veracity inference. In § 2.3, we will see a way of overcoming the requirement for the true label $y^*$, which is required for downstream reasoning tasks: by training an amortized veracity sampler $Q(V_Z \mid x, z) \propto R(V_Z)$ (or a low-temperature variant for approximate maximization).

**Working Hypothesis.** While LMs often struggle to generate logically sound and consistent CoTs during sampling, we hypothesize they are nevertheless capable of assigning higher probability to the joint distribution over the true answer and the veracity of a reasoning chain when the latter is closer to the ground-truth $v_z^*$. More formally, this means that we expect a negative correlation between $\mathbb{P}(V_z = v_z \mid Y = y^*, x, z)$ and the Hamming (L1) distance $|v_z - v_z^*|$.

While the global maximizer of the likelihood may not always coincide exactly with $v_z^*$, moving towards higher likelihoods should, on average, steer $v_z$ toward the true assignment $v_z^*$ and thereby reveal which statements in $z$ are correct. We empirically validate this hypothesis in Appendix C.5.

**Single-Bit Metropolis Updates with Simulated Annealing.** At iteration $t \in \{1, 2, \ldots, \}$ we take the current vector $v_z^{(t)}$ and perform a *single-coordinate* update:

1. Draw an index $j \sim \text{Unif}\{1, \ldots, N\}$, and propose the veracity vector obtained by flipping the $j$-th bit: $v_z' = v_z^{(t)} \oplus e_j$ (e.g. if $j = 3$ and $v_z^{(t)} = [1, 0, 1]$, then $v_z' = [1, 0, \mathbf{0}]$).
2. Accept the proposal $v_z'$ with probability $\alpha_t = \min\left\{1, \left[R(v_z')/R(v_z^{(t)})\right]^{\beta_t}\right\}$, where inverse temperature $\beta_t$ is set to 1 in the basic Metropolis algorithm. We also experiment with a schedule for $\beta_t$, $0 < \beta_0 < \cdots < \beta_M$, in the style of simulated annealing, to balance the exploration and exploitation tradeoff.
3. Set $v_z^{(t+1)} = v_z'$ if accepted, otherwise let $v_z^{(t+1)} = v_z^{(t)}$.

Because the proposal flips exactly one bit and is symmetric, the acceptance ratio simplifies to the likelihood ratio.

**Greedy–Tree Initialization.** Before running the stochastic search, we heuristically pick an initial vector $v_z^{(0)}$ using a depth-first greedy procedure. For $i = 1$ to $N$:

1. Construct partial candidates that agree on positions $1{:}i{-}1$ and set $v_{z_i} = 0$ or 1, leaving $i{+}1{:}N$ unassigned.
2. Query the model for the partial score $\tilde{R}(v_{1:i}) = P_{\text{LM}}(x \ z_{1:i} \ v_{1:i} \ y^*)$, letting the LM internally marginalize the unspecified bits.
3. Fix $v_{z_i}^{(0)}$ to the value that yields the larger $\tilde{R}$.

This greedy sequential "tree" search provides a warm start that is often close to a high-reward basin, allowing the subsequent single-bit Metropolis updates to converge more quickly.

## 2.3 AMORTIZED VERACITY INFERENCE (AVI)

In the spirit of variational expectation–maximization (EM), higher-reward samples of the veracity vector $V_z$ obtained via VS (Eq. 3) are used as pseudo-labels to train the generative model $P_{\text{LM}}$ via supervised fine-tuning, yielding the amortized sampler $Q$. Initializing at $P_{\text{LM}}(V_z \mid x z)$ and fine-tuning on VS samples obtained with simulated annealing towards zero temperature results in a an approximate maximizer of the proxy-reward $Q(V_z \mid x, z) \propto \lim_{\beta \to \infty} R(V_z)^{\beta}$.

Unlike the test-time search method (VS), $Q(V_z \mid x, z)$ is trained to predict veracity without conditioning on the answer $Y$. This provides two primary benefits: (1) it enables rapid run-time inference for verifying a CoT $z$ in a zero-shot manner without test-time search and before seeing the correct answer, and (2) it has the potential to improve reasoning by serving as feedback to identify erroneous reasoning steps in need of correction or re-generation. In some tasks (like PRONTOQA), reasoning steps predicted to be incorrect by AVI can be negated (by inserting a "not" token), yielding a corrected explanation $z'$. The correction is expected to be a more accurate reasoning chain, i.e. $P_{\text{LM}}(y^* \mid x z') > P_{\text{LM}}(y^* \mid x z)$, and we empirically validate this hypothesis in § 4.4. For tasks where negation can't be used for correction, a more general strategy involves simply appending the predicted veracity label ("True" or "False") to the statement, and we showcase this approach in our COMMONSENSEQA experiments.

## 3 RELATED WORK

**Automated feedback, self-correction, and self-improvement.** Several self-correction and self-improvement methods (Pan et al., 2024) rely on critics to provide feedback to a reasoning model, with the aim of correcting a CoT or guiding its re-generation. Sources of feedback include supervised classifiers and reward models (Rajani et al., 2019; Lightman et al., 2024; Cobbe et al., 2021; Yang et al., 2022), evidence-based fact verifiers (Chern et al., 2023; Li et al., 2023; Manakul et al., 2023), and zero/few-shot prompted LMs (to provide natural language feedback, or scalar scores) (Madaan

et al., 2023; Weng et al., 2023; Xie et al., 2023; Yao et al., 2023; Shinn et al., 2023). At training time, the final-answer itself can be used to fine-tune a reasoning model to output reasoning chains that reach the correct conclusion (Zelikman et al., 2022), which may involve variational EM to learn—and do inference in—an LVM relating reasoning chains and final answers Phan et al. (2023); Hu et al. (2024). Iterative approaches based on in-context learning to teach LMs to correct rationales by contrasting them with examples of correct rationals have also been proposed Zhou et al. (2024). RL for eliciting CoT reasoning, specifically with binary feedback based on the answers produced by CoTs, has become prominent following models like o1 (OpenAI, 2024) and DeepSeek-R1 (Guo et al., 2025).

Our work can be viewed as *complementary* to self-correction and self-improvement methods, proposing a new method for identifying errors that can serve as a useful source of feedback for any self-correction/self-improvement method; alternatively, it can be useful on its own for monitoring purposes. In contrast to standard feedback obtained via zero- or few-shot prompting of an LM, which has been found to be fragile (in part due to prompt sensitivity) and degrade performance in certain cases (Huang et al., 2023), our proposed method samples directly from the posterior distribution of interest in an LVM relating stepwise veracity to final answers. It does not require examples of correct reasoning chains, and the required instruction-prompt is minimal. Decomposing self-correction into verification and refinement sub-tasks has also been proposed in Zhang et al. (2024c), outside the context of LVMs.

**Process reward models (PRMs).** PRMs score the quality of intermediate reasoning steps and are widely used in LM post-training. They differ from our method in two key ways: (i) typically, PRMs require step-level ground-truth labels (e.g. provided by humans (Lightman et al., 2024)); (ii) supervised/reinforcement-learning fine-tuning methods to train PRMs using only outcome-supervision (final-answer labels) learn stepwise rewards that represent value/advantage and capture instrumental utility rather than stepwise correctness, and can reward useful-but-incorrect steps (Uesato et al., 2022; Zha et al., 2025; Yuan et al., 2025; Cui et al., 2025). In contrast, VS and AVI explicitly target step *veracity* without the need for process supervision, by approximating latent-veracity inference in an LVM.

**Search-based inference.** Search-based inference complements prompt engineering and fine-tuning to enhance reasoning in LMs. Simple approaches utilize CoT resampling and majority-voting to achieve improvements over single-pass generation (Wang et al., 2023; Xue et al., 2023). *Best-of-N* methods extend the sampled candidate pool and re-ranks outputs via specialized ranking models (Cobbe et al., 2021; Snell et al., 2025); however, these methods scale linearly with the number of candidates and do not revise intermediate reasoning steps. Others frame reasoning as a combinatorial search problem, exemplified by tree search prompting, which explores and prunes reasoning branches through learned or heuristic value functions (Yao et al., 2023; Xie et al., 2023), its extensions involving differentiable relaxations (Xu et al., 2025), amortized inference with GFlowNets (Bengio et al., 2021; Yu et al., 2024), and Monte-Carlo Tree Search (Luo et al., 2024; Zhang et al., 2024a; Xie et al., 2024). Our VS method is based on local search (MCMC), which is efficient in the lower-dimensional search space of veracity, and it borrows several ideas from the related work cited above.

## 4 EXPERIMENTS

In this section we evaluate VS emperically,[1] validating our hypothesis that using $P_{LM}(v_z \ y^* \mid x \ z)$ as a proxy reward improves the correctness of the veracity vector (with respect to the ground-truth $v_z^*$).

**Benchmarks.** First, we use the PRONTOQA benchmark (Saparov & He, 2022), because it gives us the ability to synthesize correct reasoning chains ($z^*$) synthetically, facilitating the introduction of controlled errors by corrupting specific steps, resulting in an incorrect proof $\tilde{z}$. Key advantages of PRONTOQA include binary veracity labels for each logical deduction steps, and the ability to generate proofs from a fictional ontology to isolate the LM's logical reasoning capability from it's ability to retrieve memorized facts/trends acquired during training. Moreover, PRONTOQA allows for adjusting the reasoning chain length to test our method's scalability to more complex scenarios. Finally, we use PRONTOQA to assess AVI and its impact on answer prediction ($y^*$).

---

[1]Source code: `https://github.com/alstn12088/veracity_inference`

Table 1: Mean Hamming Similarity (± std) on PRONTOQA, GSM8K, and COMMONSENSEQA (1,000 examples each).

| Dataset | Method | Qwen-4B | Qwen-8B | Llama-3B | Llama-8B |
|---|---|---|---|---|---|
| PRONTOQA | Recursive | 0.691 ± 0.167 | 0.667 ± 0.139 | 0.538 ± 0.117 | 0.471 ± 0.057 |
| | Many2Many | 0.590 ± 0.155 | 0.683 ± 0.142 | 0.506 ± 0.161 | 0.530 ± 0.157 |
| | Voting | 0.603 ± 0.152 | 0.692 ± 0.138 | 0.514 ± 0.156 | 0.536 ± 0.153 |
| | CoT | 0.591 ± 0.201 | 0.384 ± 0.201 | 0.459 ± 0.041 | 0.515 ± 0.162 |
| | VS (ours) | **0.910** ± 0.118 | **0.945** ± 0.096 | **0.948** ± 0.072 | **0.964** ± 0.072 |
| GSM8K | Recursive | 0.540 ± 0.167 | 0.617 ± 0.136 | 0.568 ± 0.096 | 0.568 ± 0.096 |
| | Many2Many | 0.620 ± 0.126 | 0.650 ± 0.139 | 0.566 ± 0.096 | 0.567 ± 0.096 |
| | Voting | 0.623 ± 0.127 | 0.654 ± 0.138 | 0.566 ± 0.096 | 0.553 ± 0.128 |
| | CoT | 0.614 ± 0.166 | 0.695 ± 0.204 | 0.496 ± 0.164 | 0.496 ± 0.165 |
| | VS (ours) | **0.711** ± 0.155 | **0.751** ± 0.193 | **0.614** ± 0.143 | **0.646** ± 0.157 |
| COMMONSENSEQA | Recursive | 0.607 ± 0.217 | 0.509 ± 0.220 | 0.505 ± 0.219 | 0.506 ± 0.219 |
| | Many2Many | 0.517 ± 0.227 | 0.534 ± 0.212 | 0.504 ± 0.220 | 0.503 ± 0.219 |
| | Voting | 0.521 ± 0.226 | 0.533 ± 0.208 | 0.504 ± 0.220 | 0.505 ± 0.219 |
| | CoT | 0.695 ± 0.230 | 0.590 ± 0.220 | 0.507 ± 0.219 | 0.535 ± 0.227 |
| | VS (ours) | **0.935** ± 0.123 | **0.931** ± 0.119 | **0.836** ± 0.176 | **0.903** ± 0.137 |

We additionally evaluate our method across other reasoning domains, namely mathematical reasoning (Cobbe et al., 2021, GSM8K) and commonsense reasoning (Talmor et al., 2018, COMMONSENSEQA). In contrast to PRONTOQA, these datasets do not provide us with the ability to generate corrupted reasoning chains in a controlled manner, limiting our ability to measure the accuracy of veracity assignments. We overcome this by generating structured reasoning chains via a more powerful oracle model (GPT-4.1) using OpenAI's API for generating structured outputs (a sequence of strings, each of which corresponds to a reasoning steps), conditioned on the correct answer $y^*$. We treat the generated reasoning chains as ground truth, assuming $v_z^* = 1$, and then corrupt them with controlled perturbations.

**Base LMs.** We evaluate our approach using several representative LMs: Qwen 3 (4B), Qwen 3 (8B), Llama 3.2 (3B), and Llama 3 (8B).

**Baselines.** Our primary goal is to demonstrate the effectiveness of using the joint probability $P_{\text{LM}}(v_z \, y^* \mid x \, z)$ as a proxy reward. We introduce autoregressive baselines that generate $V_z$ in a tractable manner by modifying the decoding trajectory from the original intractable form ($X \rightarrow Z \rightarrow V_z \rightarrow Y$) to the tractable form ($X \rightarrow Y \rightarrow Z \rightarrow V_z$). These baseline methods directly query the LM to generate $V_z$ given $x$, $y$, and $z$, using few-shot examples.

These baseline inference methods are further categorized as follows: (1) Many2Many Inference (**Many2Many**): Generates a complete veracity vector $v_z$ in one-shot for a given sequence of reasoning steps $z$; (2) Many2Many Inference with the addition of CoT-prompting (**CoT**): Given $x$, $y$, and $z$, we prompt the model to generate a CoT before outputting the value of $V_z$, to explain its reasoning; (3) One-Shot Majority Voting (**Voting**): Uses majority voting across $M = 50$ samples generated at a higher temperature ($T = 0.5$) to predict the veracity assignment, whereas other methods employ a greedy-like temperature ($T = 0.01$) to enhance consistency; (4) Recursive Inference (**Recursive**): Predicts each stepwise veracity label $v_{z_i}$ recursively, where $i$ corresponds to the index of the statement $i$ in the reasoning chain, conditioned on previously inferred labels $v_{z_{1:i-1}}$ and corresponding statement identities $z_{1:i-1}$. The generative trajectory is thus structured as $X \rightarrow Y \rightarrow Z_1 \rightarrow V_{z_1} \rightarrow \cdots \rightarrow Z_N \rightarrow V_{z_N}$. We provide the same five few-shot demonstrations for all baselines, including our proposed method.

Details pertaining to implementation, CoT corruption, and prompting, are provided in Appendix A.

## 4.1 EVALUATING THE ACCURACY OF VERACITY INFERENCE

To quantify the performance of VS, we compute the Hamming similarity between the predicted correctness vector $v_z$ and the ground-truth vector $v_z^*$: $\text{Sim}(v_z, v_z^*) = 1 - \|v_z - v_z^*\|_1 / L$, where $\|\cdot\|_1$

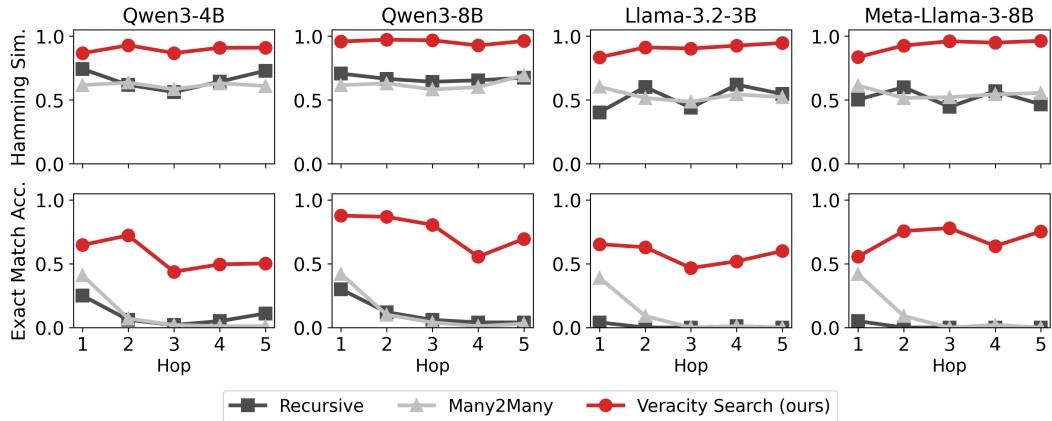

Figure 2: Veracity inference evaluation for different number of hops in PRONTOQA. **Top row**: Mean Hamming Similarity; **Bottom row**: Mean Exact Match Accuracy. Mean is estimated using 100 test samples.

denotes the element-wise $\ell_1$-norm and $L = |v_z| = |v_z^*|$ is the length of the vector. A similarity of 1 indicates perfect agreement, whereas 0 signifies complete disagreement.

The results for PRONTOQA, GSM8K, and COMMONSENSEQA are presented in Table 1. In both tasks, VS (run for 200 iterations on the proxy-reward Eq. 3), consistently outperforms the other baselines. In particular, VS attains near-perfect identification accuracy on PRONTOQA.

In the GSM8K mathematical reasoning task, VS consistently outperformed baseline methods across all tested LMs. The remaining gap with ground-truth veracity labels is likely due to the distribution of errors induced by our approach of corrupting CoTs with noise. In cases where a reasoning step can be viewed as a Boolean assertion (PRONTOQA and COMMONSENSEQA), flagging errors is no different from correcting them. In the setting of GSM8K, however, beyond flagging arithmetic errors, one also needs to re-do the flawed calculations in order to correct the answer or identify additional errors, impacting both the baselines and VS.

The importance of the CoT corruption scheme also highlights the need to evaluate VS under other distributions of erroneous reasoning chains. We conduct such an evaluation on PRONTOQA and COMMONSENSEQA in Appendix C.1, and find that VS outperforms baselines in a wider variety of settings. Finally, we evaluate VS in cases where veracity is a categorical variable with more than two classes in Appendix C.4, and assess the sensitivity of VS accuracy to model size in Appendix C.7.

## 4.2 SCALING TO LONGER REASONING CHAINS

PRONTOQA structures each reasoning trace into discrete hops: one hop corresponds to the application of a deduction rule (modus ponens), which itself is broken down in more than one statement. For example, a 5-hop can involve up to 13 reasoning steps. The number of hops allows us to control the length of the reasoning chain and therefore the complexity of inference.

We evaluated four LMs (Qwen3-4B, Qwen3-8B, Llama-3.2-3B, Llama-3-8B) on logical reasoning problems ranging from 1 to 5 hops, uniformly flipping half of the ground-truth statements in the ground-truth CoT $z^*$ to produce corrupted versions $\tilde{z}$. Fig. 2 shows how well each method recovers the ground-truth veracity vector. In addition to Hamming Similarity, we also measured Exact Match Accuracy, which is equal to 1 if the predicted $v_z$ matches the ground truth $v_z^*$, and 0 otherwise.

All methods maintained nearly constant Hamming similarity throughout the range of hops, with VS consistently above 0.85, outperforming baselines by 20–25 points. Exact-match accuracy inevitably decayed with increased hops, as the probability of correctly predicting all errors shrinks exponentially in $|z|$. Nevertheless, VS maintained relatively stable Exact-match accuracy even in 5-hop scenarios where the baselines already demonstrate a significant performance gap, failing to identify any error.

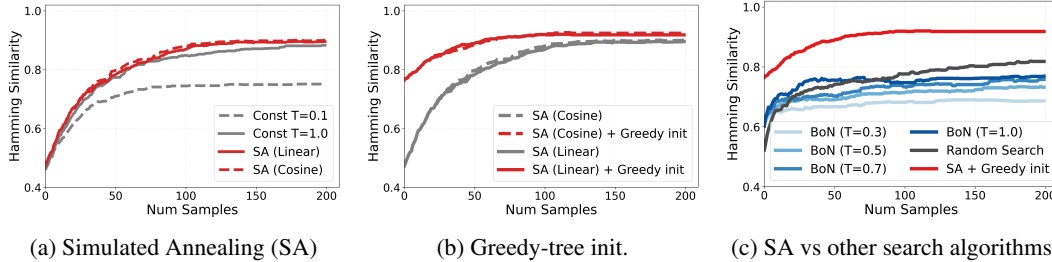

(a) Simulated Annealing (SA)  (b) Greedy-tree init.  (c) SA vs other search algorithms

Figure 3: Ablation study for search hyperparameters. SA: Simulated Annealing; Const: Constant Temperature; BoN($T$): Best-of-$N$ baseline using LM proposals sampled at temperature $T$. Mean value is estimated over 100 test samples from PRONTOQA.

Table 2: Hamming similarity between predicted veracity from the AVI and ground truth labels $v_z^*$. Mean similarity is computed over 100 test samples from 3,4, and 5-hop PRONTOQA.

| Base LLM | Qwen 4B | | | Qwen 8B | | |
|---|---|---|---|---|---|---|
| | 3-hop | 4-hop | 5-hop | 3-hop | 4-hop | 5-hop |
| Many2Many | 0.710±0.197 | 0.779±0.131 | 0.684±0.131 | 0.663±0.146 | 0.643±0.223 | 0.665±0.176 |
| AVI (ours) | **0.886**±0.143 | **0.921**±0.010 | **0.913**±0.108 | **0.956**±0.008 | **0.967**±0.069 | **0.955**±0.081 |

## 4.3 ABLATION STUDY FOR SEARCH HYPERPARAMETERS

We conduct three ablations to analyze key design choices in VS, averaging results over identical PRONTOQA splits (see Figure 3): (1) **Simulated annealing** (Figure 3a): Linear and cosine annealing ($T_1 = 1/\beta_! = 2, T_M = 1/\beta_M = 0.1$) yield similar results and slightly outperform constant temperature ($T = 1$), demonstrating the advantage of gradually increasing the inverse temperature to escape local optima for improved approximate global maximization of the proxy-reward. A constant inverse temperature ($T = 0.1$) results in getting stuck in local optima; (2) **Greedy-tree initialization** (Figure 3b): The greedy initialization method using simple tree search (§ 2.2) significantly boosts sample efficiency by starting from a high-quality initial solution; (3) **Comparison with other search algorithms** (Figure 3c): Our Metropolis algorithm with Simulated Annealing clearly outperforms uniform random bit-flips (random search) and Best-of-$N$ (BoN) LM-generated proposals. Random search wastes resources, while BoN lacks local exploration, underscoring the necessity of structure-aware local moves.

Collectively, these ablations confirm that simulated annealing, principled initialization, and structured local moves significantly enhance the performance of VS. In Appendix C.3, we extend our method to block Metropolis, allowing for multi-bit flips in settings where more complex joint dependencies exist between veracity variables.

## 4.4 EVALUATION OF AVI

Qwen3-4B and Qwen3-8B models were fine-tuned using pseudo-labels generated by VS, for 5,000 contexts (($x, y^*, z$) tuples) in a 4-hop PRONTOQA training dataset, as described in Appendix A.

**Veracity inference.** Table 2 reports the Hamming similarity between $v_z$ selected by the AVI and the ground-truth veracity vector $v_z^*$ in a test set of 100 PRONTOQA examples. Despite being fine-tuned only on 4-hop proofs, the model generalizes to unseen chain lengths. Across both Qwen backbones, Hamming similarity increases by ≈15–25 points relative to the strongest one-shot baseline, closing most of the gap to the optimal similarity of 1.0.

**Effect on downstream reasoning.** In PRONTOQA, we first generate correct proofs $z^*$, then inject errors by randomly negating a subset of statements to create a corrupted reasoning chain $\tilde{z}$. When an LM conditions directly on this flawed $\tilde{z}$, its ability to predict the correct answer $y^*$ drops sharply. A simple self-correction method leveraging AVI works as follows: first, incorrect statements are identified by the AVI, then these statements are negated to form a corrected chain $z'$ (e.g. replacing "*every impus is temperate*" with "*not every impus is temperate*"). This process is illustrated in Fig. 1.

Table 3: Reasoning accuracy for inferring $y^*$ from PRONTOQA problems given synthetic noisy chains. Average accuracy and standard deviation computed over 100 problems.

| Method | Qwen 4B | | | Qwen 8B | | |
|---|---|---|---|---|---|---|
| | 3-hop | 4-hop | 5-hop | 3-hop | 4-hop | 5-hop |
| No Correction | 0.60±0.05 | 0.52±0.05 | 0.59±0.05 | 0.54±0.05 | 0.65±0.05 | 0.52±0.05 |
| Self Correction | 0.54±0.05 | 0.60±0.05 | 0.48±0.05 | 0.54±0.05 | 0.58±0.05 | 0.46±0.05 |
| AVI (ours) | **0.68±0.05** | **0.72±0.04** | **0.77±0.04** | **0.87±0.03** | **0.85±0.04** | **0.81±0.04** |

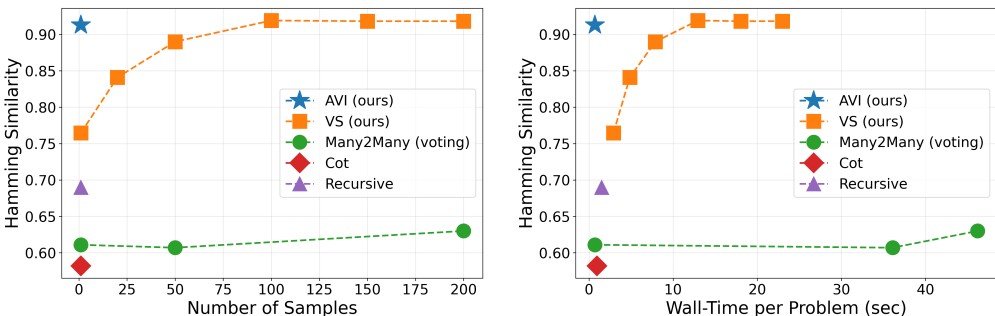

Figure 4: Test-time inference efficiency of Qwen3-4B on 100 problems $(x, z, y^*)$ from 5-hop PRONTOQA for inferring $v_z$. Inference wall time is shown as the number of samples per problem scales.

Table 3 compares three scenarios: **(a)** using the uncorrected $\tilde{z}$, **(b)** self-correction using the Many2Many baseline, and **(c)** employing AVI for self-correction. Conditioning the LM on $z'$ boosts the conditional probability of the true answer by up to 25% on Qwen-8B and by 10–12 points on Qwen-4B. The improvement is consistent across 3-, 4-, and 5-hop proofs, suggesting that simply correcting mistakes in a CoT can result in improved reasoning accuracy. We extend this evaluation to LM-generated reasoning chains $z$ (with no synthetic corruption) in Appendix C.2.

Inference-time self-improvement methods are related to self-correction, and aim to improve reasoning through resampling and editing, occasionally utilising a distinct feedback module for guiding correction. Our approach is complementary to these self-improvement methods, as it focuses on identifying statements that need correction, and would therefore fit naturally as a feedback module. In Appendix C.2, we illustrate how AVI can be used as the feedback module in the self-improvment method Self-Refine (Madaan et al., 2023), replacing the few-shot prompted LM (corresponding to our baseline Many2Many) that is typically used for this purpose. We find that AVI continues to outperform baselines for this type of application, but note that the magnitude of improvement in reasoning performance is smaller than the gain observed in terms of veracity inference accuracy in the preceding experiments. This suggests that verification alone is not the only bottleneck in self-correction/improvement frameworks (Zhang et al., 2024c), and more work is needed to better understand how to make the best use of a stronger verification signal to guide CoT-(re)generation.

### 4.5 SAMPLE EFFICIENCY AND WALL-TIME

Finally, we evaluate the sample-efficiency and inference-time requirements of our proposed method compared to the same baselines from previous experiments. We use 100 examples from the 5-hop PRONTOQA dataset and run all experiments on the same NVIDIA RTX 6000 ADA (48GB) GPU.

As depicted in Figure 4 (left), VS outperforms all baseline methods in terms of sample efficiency, even when limited to a single sample using greedy tree-based initialization. Performance scales favorably, reaching optimal results at around 100 samples. The zero-shot (1-sample) AVI provides Pareto-optimal performance, comparable to the 100-sample iteration of VS. As shown in Figure 4 (right), VS also uses less inference time per sample compared to other baselines. This efficiency stems from the parallelized reward computations using LM likelihoods evaluated on sampled sequences, eliminating the need for sequential token-level decoding that other baselines require. We include a complementary analysis of the inference-time cost of our approach in Appendix C.6.

## 5 CONCLUSION

We introduced a framework for stepwise error-identification that extends the latent variable representation of a CoT by disentangling its identity $z$ from its veracity $V_z$. We proposed a discrete search algorithm, Veracity Search (VS), to efficiently search over boolean veracity vectors, and proposed a method (Amortized Veracity Inference; AVI) for fine-tuning an LM with pseudo-labels from VS, allowing us to apply VS in contexts where the final answer is unknown. Empirically, we found that VS outperforms in-context learning baselines in logical, mathematical, and commonsense reasoning tasks when assessing the accuracy of the predicted veracity of individual reasoning steps, and that AVI can complement downstream tasks that utilize veracity labels as feedback for correction or for guiding the re-generation of erroneous reasoning chains towards ones that increase reasoning performance.

There are important limitations to our work. The performance of VS can be sensitive to the distribution of errors, and most of our analysis pertains to artificially corrupted CoTs. Preliminary results suggest that these techniques also work for naturally occurring errors (§C.1), but these claims should be tested in a broader range of dedicated experiments. Ultimately, important applications of error-identification (such as self-correction) often require more than flagging mistakes: inferring veracity $v_z$ of a CoT does not render the identity $z$ of the reasoning steps unimportant. In the appendix (§C.2) we prototype a way of integrating AVI with self-improvement methods to resample erroneous statements in $z$, but designing more powerful reasoning systems that combine veracity inference with iterative test-time correction schemes remains an important avenue for future work. Our method could also provide a new training signal for reasoning models, for example by using EM to jointly update the generative model with veracity samples obtained with AVI, or by providing a veracity-specific process-level training signal for PRMs in label-scarcity scenarios.

## ACKNOWLEDGMENTS

We thank researchers at Mila (Joumana Ghosn, Pierre-Luc St-Charles, Marc-Antoine Rondeau, Mohsin Hasan, Siddarth Venkatraman, Yaroslav Kivva) and KAIST (Junyeop Baek, Doojin Baek) for insightful discussions and assistance with this project. This research is supported by the Canadian AI Safety Institute Research Program at CIFAR through a Catalyst award. The research was enabled in part by computational resources provided by the Digital Research Alliance of Canada (`https://alliancecan.ca`), Mila (`https://mila.quebec`), and NVIDIA. The authors acknowledge funding from CIFAR, NSERC and the Future of Life Institute. Minsu Kim acknowledges funding from KAIST Jang Young Sil Fellow Program. Jean-Pierre Falet is supported by a Doctoral Vanier Canada Graduate Scholarship (FRN: CGV-192746). Moksh Jain acknowledges funding from a FRQNT Doctoral Fellowship (`https://doi.org/10.69777/366694`). Minsu Kim, Sungsoo Ahn, and Sungjin Ahn were supported by GRDC Cooperative Hub Program (RS-2024-00436165) through the National Research Foundation of Korea (NRF).

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

# A  DETAILED IMPLEMENTATIONS AND EXPERIMENTS ON BASELINES

## A.1  DATASETS

**PRONTOQA.** We generate $(x, y, z)$ triples with the official PRONTOQA synthesiser, which samples first–order–logic templates with a fictional ontology. For every instance we uniformly draw a reasoning depth $d \in \{1, \ldots, 5\}$ ("hops"). Each ground-truth veracity vector $v_z^* \in \{0, 1\}^{|z|}$ is corrupted into $v_z$ by independently negating every entry with probability 0.5.

**GSM8K.** We retain the original question–answer pairs $(x, y, z)$ and obtain reference chains of thought $z^*$ by querying GPT-4.1 with $T = 0.1$ under a deterministic JSON schema. Corruption is applied by perturbing numerical constants in $z^*$: each integer is randomly shifted by ±1, doubled, or halved so that exactly 50% of all statements become incorrect.

**COMMONSENSEQA.** Reasoning chains $z^*$ are generated similarly to GSM8K, by prompting GPT-4.1 to produce structured explanations conditioned on the correct answer $y^*$. Corruption is applied as in PRONTOQA, where each statement in $z^*$ is independently negated with probability 0.5, yielding corrupted chains $z$ with associated veracity labels $v_z$.

## A.2  BASELINE INFERENCE METHODS

All baselines bypass the intractable problem of in-filling $V_z$ in the sequence $X \to Z \to (V_z) \to Y$ by querying for the probability of $v_z$ after observing $(x, y, z)$. Unless stated otherwise, the sampling temperature is $T = 0.01$.

- **Many2Many**: one-shot prediction of the full vector $v_z$.
- **Many2Many+CoT**: generates an intermediate rationale $R_{V_z}$ before emitting $v_z$.
- **Majority Voting**: draws $M = 50$ samples at $T = 0.5$ and returns the element-wise majority.
- **Recursive**: predicts labels sequentially, conditioning on past $\langle z_i, v_{z_i} \rangle$ pairs.

We provide the prompt templates below:

**Logical reasoning and commonsense reasoning.**

```
(... few shot demos with intruction)
### Context
{X}

### Query
{logical_question which is last sentense of X}

### Answer
{Y}

### Explanation Steps
Step 1: {Z_1}
...
Step N: {Z_N}

Give your judgement in JSON:
{"Label": [true, false, ...]}
```

**Mathematical reasoning.**

```
(... few shot demos with intruction)
### Problem
{X}

### Answer
{Y}

### Solution Steps
Step 1: {Z_1}
...
Step N: {Z_N}

Give your judgement in JSON:
{"Label": [true/false, ...]}
```

**Prompt Example (Logic, PRONTOQA).**

This is a prompt example for Many2Many inference:

```
(... few shot demos with intruction)
### Context
Jompuses are overcast. Jompuses are yumpuses. Every yumpus is an impus.
Yumpuses are wooden. Lempuses are jompuses. Each impus is a gorpus.
Gorpuses are not transparent. Grimpuses are not sweet. Each impus is not nervous.
Jompuses are tumpuses. Zumpuses are brimpuses. Dumpuses are not dull.
Every lempus is a dumpus. Every numpus is not overcast. Each zumpus is orange.
Each impus is a shumpus. Every lempus is slow. Every tumpus is discordant.
Yumpuses are grimpuses. Polly is a jompus. Polly is a zumpus.

### Query
True or false: Polly is overcast.

### Answer
True

### Explanation Steps
Step 1: Polly is a jompus.
Step 2: Jompuses are overcast.
Step 3: Polly is overcast.

Give your judgement in JSON:
{"Label": [false, true, false]}
```

This is a prompt example for the recursive inference:

```
(... few shot demos with intruction)
### Context
Jompuses are overcast. Jompuses are yumpuses. Every yumpus is an impus.
Yumpuses are wooden. Lempuses are jompuses. Each impus is a gorpus.
Gorpuses are not transparent. Grimpuses are not sweet. Each impus is not nervous.
Jompuses are tumpuses. Zumpuses are brimpuses. Dumpuses are not dull.
Every lempus is a dumpus. Every numpus is not overcast. Each zumpus is orange.
Each impus is a shumpus. Every lempus is slow. Every tumpus is discordant.
Yumpuses are grimpuses. Polly is a jompus. Polly is a zumpus.

### Query
True or false: Polly is overcast.

### Answer
True

### Explanation Steps (with labels so far)
Step 1: Polly is a jompus.  Label: true (self verified)
...
Step k-1: Jompuses are overcast.  Label: false (self verified)
Step k:  Polly is overcast.  Label:

Predict the label for last only.
Return JSON: {"Label": true|false}
```

These prompts are recursively queried for $k = 1, \ldots, N$.

**Prompt Example (Commonsense, COMMONSENSEQA).**

This is a prompt example for Many2Many inference:

```
(... few shot demos with instruction)
### Question
Where do you find wild cats?

### Query
A) trouble, B) dog's mouth, C) nature, D) floor, E) warm place.

### Answer
C

### Explanation Steps
Step 1: The question asks where wild cats are found.
Step 2: Option C is 'nature'.
Step 3: Wild cats are animals that live in natural environments.
Step 4: Nature refers to the outdoors and natural habitats.
Step 5: Therefore, Option C (nature) correctly answers where wild cats are found.

Give your judgement in JSON:
{"Label": [true, false, true, true, false]}
```

This is a prompt example for Recursive inference:

```
(... few shot demos with instruction)
### Question
Where do you find wild cats?

### Query
A) trouble, B) dog's mouth, C) nature, D) floor, E) warm place.

### Answer
C

### Explanation Steps (with labels so far)
Step 1: The question asks where wild cats are found. Label: true
Step 2: Option C is 'nature'. Label: false
Step 3: Wild cats are animals that live in natural environments. Label: false
Step 4: Nature refers to the outdoors and natural habitats. Label: true
Step 5: Therefore, Option C (nature) correctly answers where wild cats are found. Label:

Predict the label for last only.
Return JSON: {"Label": true|false}
```

These prompts are recursively queried for $k = 1, \ldots, N$.

**Prompt Example (Math, GSM8K).**

This is a prompt example for Many2Many inference:

```
(... few shot demos with intruction)
### Problem
Janet's ducks lay 16 eggs per day. She eats three for breakfast every morning
and bakes muffins for her friends every day with four. She sells the remainder
at the farmers' market daily for $2 per fresh duck egg. How much in dollars
does she make every day at the farmers' market?

### Answer
18

### Solution Steps
Step 1: Determine the total number of eggs laid by Janet's ducks each day.
        Intermediate output: 16
Step 2: Subtract the number of eggs Janet eats for breakfast from the total eggs.
        Intermediate output: 13
Step 3: Subtract the number of eggs used for baking muffins from the remaining eggs.
        Intermediate output: 9
Step 4: Identify the number of eggs Janet has left to sell at the farmers' market.
        Intermediate output: 9
Step 5: Determine the price per egg Janet sells at the market.
        Intermediate output: 2
Step 6: Multiply the number of eggs Janet sells by the price per egg to find her earnings.
        Intermediate output: 18
Step 7: Verify that 9 eggs multiplied by $2 per egg equals $18.
        Intermediate output: 18

Give your judgement in JSON:
{"Label": [false, false, false, true, false, true, true]}
```

This is a prompt example for Recursive inference:

```
(... few shot demos with intruction)
### Problem
Janet's ducks lay 16 eggs per day. She eats three for breakfast every morning
and bakes muffins for her friends every day with four. She sells the remainder
at the farmers' market daily for $2 per fresh duck egg. How much in dollars
does she make every day at the farmers' market?

### Answer
18

### Solution Steps (with labels so far)
Step 1: Determine the total number of eggs laid by Janet's ducks each day.
        Intermediate output: 16. Label: true
...
Step k: Identify the number of eggs Janet has left to sell at the farmers' market.
        Intermediate output: 9. Label: true
Step k+1: Determine the price per egg Janet sells at the market.
        Intermediate output: 2. Label:

Predict the label for last only.
Return JSON: {"Label": true|false}
```

## A.3 VERACITY SEARCH (VS)

VS keeps the original trajectory $X \rightarrow Z \rightarrow V_z \rightarrow Y$ and optimises $V_z$ with a Metropolis algorithm that leverages the latent variable model-based posterior $R(V_z) = P_{\text{LM}}(Y, V_z \mid X, Z)$ as a reward. In this section, we describe the detailed setting and prompt of our method.

**Iterations** 200

**Temperature** linear annealing from $T_0 = 2.0$ to $T_{200} = 1.0$

**Proposal** single-bit flip (Hamming-1)

**Reward Prompt (Logic and Commonsense).**

```
(... few shot demos with instruction)
Label **True** if the step follows from Context, **False** otherwise.

### Context
{X}

### Query
{logical_question (last sentense of X)}

### Explanation Steps
{Z_1 ... Z_N}

### Label Vector (V_z)
{0/1 sequence}

### Answer
{Y}
```

**Reward Prompt (Math).**

```
(... few shot demos with instruction)
Think step by step. Mark each step as **Correct** or **Incorrect**.

### Question
{X}

### Solution Steps
{Z_1 ... Z_N}

### Answer
{Y}
```

### A.4 Amortized Veracity Inference (AVI)

**Training.** We amortize VS into a lightweight veracity inference machine by fine-tuning Qwen3-4B/8B on 5,000 labeled contexts produced by the VS.

- **LoRA** rank 8, $\alpha = 32$
- **Batch / Accum.** 32 / 8
- **Optimizer / LR** AdamW (Loshchilov & Hutter, 2019), $1 \times 10^{-4}$
- **Hardware** single NVIDIA A100L (80GB)
- **Runtime** 16 min (4B) / 24 min (8B) per epoch

Validation accuracy saturates after a single epoch (Fig. 5).

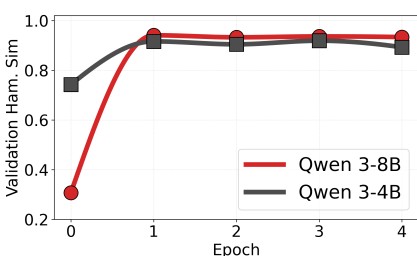

Figure 5: Validation curve.

**Inference for $V_z$.** We do Many2Many inference without accessing true answer $Y$. The prompt template is as follows:

```
(... few shot demos with intruction)
### Context
{X}

### Query
{logical_question which is last sentense of X}

### Answer
Unknown

### Explanation Steps
Step 1: {Z_1}
...
Step N: {Z_N}

Give your judgement in JSON:
{"Label": [true, false, ...]}
```

**Inference for $Y$.** Using corrected $Z$ with $V_z$ by negating for False statement, we can infer $Y$. The prompt template is as follows:

```
(... few shot demos with intruction)
### Context
{X}

### Query
{logical_question which is last sentense of X}

### Explanation Steps
Step 1: {Z_1}
...
Step N: {Z_N}

Your answer:
```

## B NATURAL LANGUAGE EXAMPLES

We provide 3 examples (first 3 examples in a batch) for logical and math reasoning correction using VS.

**Logic example 1: Max is not wooden.**

```
### Context (X)
Numpuses are not wooden. Vumpuses are lempuses. Rompuses are not dull.
    ...
Max is a vumpus. Max is a lorpus.

### Query (X)
True or false: Max is not wooden.

### Answer (Y)
True

### Corrupted chain (Z)
1. Max is not a vumpus.
2. Each vumpus is not a brimpus.
3. Max is a brimpus.
4. not Brimpuses are grimpuses.
5. Max is not a grimpus.
6. Grimpuses are shumpuses.
7. Max is a shumpus.
8. Shumpuses are numpuses.
9. Max is a numpus.
10. Numpuses are wooden.
11. Max is wooden.

### V_z pred
[F, F, T, F, F, T, T, T, T, T, T]

### V_z ground truth
[F, F, T, F, F, T, T, T, T, F, F]
```

**Logic example 2: Fae is not discordant.**

```
### Context (X)
Every rompus is not sweet. ... Fae is a sterpus. Fae is a brimpus.

### Query (X)
True or false: Fae is not discordant.

### Answer (Y)
False

### Corrupted chain (Z)
1. Fae is a brimpus.
2. Brimpuses are rompuses.
3. Fae is not a rompus.
4. Rompuses are lempuses.
5. Fae is not a lempus.
6. Each lempus is not a grimpus.
7. Fae is not a grimpus.
8. Each grimpus is a lorpus.
9. Fae is a lorpus.
10. Each lorpus is not discordant.
11. Fae is not discordant.

### V_z pred
[T, T, F, T, F, F, F, T, T, F, F]

### V_z ground truth
[T, T, F, T, F, F, F, T, T, F, F]
```

**Logic example 3: Stella is liquid.**

```
### Context (X)
Every sterpus is not melodic. ... Stella is a rompus. Stella is a dumpus.

### Query (X)
True or false: Stella is liquid.

### Answer (Y)
True

### Corrupted chain (Z)
1. Stella is a dumpus.
2. Each dumpus is a shumpus.
3. Stella is not a shumpus.
4. not Shumpuses are sterpuses.
5. Stella is a sterpus.
6. not Sterpuses are gorpuses.
7. Stella is not a gorpus.
8. Each gorpus is a jompus.
9. Stella is a jompus.
10. not Jompuses are liquid.
11. Stella is not liquid.

### V_pred
[T, T, F, F, F, F, T, T, T, F, F]

### V_z ground truth
[T, T, F, F, T, F, F, T, T, F, F]
```

**Commonsense example 1: Salad.**

```
### Question (X)
What meal usually has carrots and mostly leafy vegetables?

### Choices (Y)
A) cellar, B) salad, C) store, D) refrigerator, E) soup.

### Answer (T_Y)
B

### Corrupted chain (Z)
1. The question asks for a meal that usually has not carrots and mostly
    leafy vegetables.
2. Option B is not 'salad'.
3. A salad is not a meal that typically consists of mostly leafy
    vegetables.
4. Carrots are not commonly included as an ingredient in salads.
5. Therefore, Option B (salad) fits the description given in the question
    .

### V_z pred
[T, F, F, F, T]

### V_z ground truth
[F, F, F, F, T]
```

**Commonsense example 2: Temperate climate.**

```
### Question (X)
What is the easiest place for a human or another animal to live?

### Choices (Y)
A) workplace, B) gravesite, C) shelter, D) interesting place, E)
    temperate climate.

### Answer (T_Y)
E

### Corrupted chain (Z)
1. Not true: The question asks for the easiest place for a human or
    another animal to live.
2. Option E is 'temperate climate'.
3. False: A temperate climate provides moderate temperatures and
    conditions suitable for most humans and animals.
4. Other options such as workplace, gravesite, shelter, or interesting
    place do not inherently provide suitable living conditions for both
    humans and animals.
5. Therefore, Option E (temperate climate) is not the correct answer.

### V_z pred
[F, T, F, T, F]

### V_z ground truth
[F, T, F, T, F]
```

**Commonsense example 3: Virginia.**

```
### Question (X)
Which state is the Blue Ridge Mountains Roanoke City in?

### Choices (Y)
A) appalachians, B) north carolina, C) virginia, D) pennsylvania, E)
    montana.

### Answer (T_Y)
C

### Corrupted chain (Z)
1. The question asks which state the Blue Ridge Mountains Roanoke City is
    not in.
2. Option C is not 'virginia'.
3. Roanoke City is not located in the state of Virginia.
4. Not true: The Blue Ridge Mountains extend through Virginia.
5. Incorrect: Therefore, Option C (virginia) correctly answers the
    question.

### V_z pred
[F, F, F, F, F]

### V_z ground truth
[F, F, F, F, F]
```

**Math example 1: Janet's duck-egg earnings.**

```
### Problem (X)
Janet's ducks lay 16 eggs per day. She eats three ... How much in dollars
does she make every day at the farmers' market?

### Answer (Y)
18

### Corrupted chain (Z)
1. Determine total eggs laid.  Out: 17
2. Subtract breakfast eggs.    Out: 12
3. Subtract muffin eggs.       Out: 10
4. Eggs left to sell.          Out: 9
5. Price per egg.              Out: 1
6. Multiply eggs x price.      Out: 18
7. Verify 9 x $2 = $18.        Out: 18

### V_z pred
[F, F, F, F, F, F, T]

### V_z ground truth
[F, F, F, T, F, T, T]
```

**Math example 2: Robe fabric bolts.**

```
### Problem
A robe takes 2 bolts of blue fiber and half that much white fiber.
How many bolts in total does it take?

### Answer
3

### Corrupted chain (Z)
1. Identify blue bolts: 2  Out: 1
2. White bolts = half.      (no out)
3. Half of 2 is 1 bolt.     Out: 1
4. Add 2 + 1.               Out: 3
5. Verify total = 3.        Out: 4

### V_z pred
[T, T, T, T, F]

### V_z ground truth
[F, T, T, T, F]
```

**Math example 3: House-flipping profit.**

```
### Problem
Josh buys a house for $80,000 and spends $50,000 on repairs,
which increase the value by 150%. How much profit did he make?

### Answer
70000

### Corrupted chain (Z)
1. Purchase price.                      Out: 80001
2. Repair cost.                         Out: 50000
3. Total investment.                    Out: 130000
4. Repairs increase value by 150%.      (no out)
5. 150% of 80k.                         Out: 119999
6. New value after repairs.             Out: 200000
7. (Erroneous) value double-count.      Out: 400000
8. Profit = 200k - 130k.                Out: 70000
9. Verification step.                   Out: 140000

### V_pred
[T, T, T, T, F, T, F, T, F]

### V_z ground truth
[F, T, T, T, F, T, F, T, F]
```

## C ADDITIONAL RESULTS

### C.1 ANALYSIS OF ROBUSTNESS TO DIFFERENT ERROR DISTRIBUTIONS

Reasoning errors exhibit complex dependencies across reasoning steps, and won't necessarily be distributed uniformly. We therefore evaluate our method under different error distributions using both synthetic CoT corruption methods (§C.1.1) and naturally-occuring errors (§C.1.2).

### C.1.1 DIFFERENT PATTERNS OF SYNTHETIC COT CORRUPTION

Mistakes can concentrate near the beginning of a CoT (e.g. when premises are misapplied), or toward the end (e.g. when incorrect conclusions are inferred from preceding assertions). To examine robustness under such conditions, we evaluate three perturbation patterns in PRONTOQA: *front-side* (errors injected in the first half of reasoning steps through negating correct statements), *uniform*, and *back-side* (second half of statements are negated). Results obtained with Qwen3-8B are shown in Table 4.

Table 4: Average Hamming Similarity and Exact-Match Accuracy across 100 samples of 5-hop PRONTOQA using Qwen3-8B under different error patterns.

| Method | Hamming Similarity | | | Exact Match | | |
|---|---|---|---|---|---|---|
| | Front-side | Uniform | Back-side | Front-side | Uniform | Back-side |
| Recursive | 0.768 | 0.679 | 0.707 | 0.260 | 0.030 | 0.030 |
| Many2Many | 0.857 | 0.678 | 0.684 | 0.000 | 0.010 | 0.000 |
| **VS** | **0.983** | **0.963** | **0.947** | **0.810** | **0.720** | **0.750** |

Across all perturbation patterns, VS consistently surpasses both Recursive and Many2Many baselines. In particular, it achieves near-perfect Hamming similarity (0.947–0.983) and markedly higher exact-match accuracy (0.720–0.810), whereas baseline accuracies remain close to zero. These findings indicate that VS is robust to different distributions of errors within the reasoning chain, underscoring its applicability to realistic settings where error patterns are diverse and not easily predictable.

### C.1.2 NATURALLY OCCURRING (LM-GENERATED) ERRORS

The main results presented in this paper are derived from controlled experiments using synthetically corrupted reasoning chains, facilitating the evaluation of the accuracy of the inferred veracity assignments $v_z$. However, a practical scenario involves an LM generating a CoT $z$ on its own, with errors arising naturally as a result of the limitations of the model's reasoning capabilities. Here, we examine whether our method generalizes to this more realistic setting.

Reasoning chains $z$ were generated using structured-decoding with a Qwen3-4B model for questions $x$ sampled from PRONTOQA and COMMONSENSEQA datasets (5,000 samples each). The resulting $(x, z)$ pairs and corresponding answer labels $y^*$ were then subjected to VS to obtain predicted veracities $\hat{v}_z \in \{\texttt{True}, \texttt{False}\}^{|z|}$.

Table 5: Veracity accuracy in naturally generated CoTs (1,000 test samples for each task).

| Method | Hamming Similarity | | Exact Match Accuracy | |
|---|---|---|---|---|
| | ProntoQA | CommonsenseQA | ProntoQA | CommonsenseQA |
| Many2Many | 0.74 | 0.59 | 0.00 | 0.00 |
| VS | **0.92** | **0.86** | **0.50** | **0.59** |

We evaluated VS on 1,000 test samples (for each dataset) in terms of veracity inference accuracy (Hamming similarity and exact match accuracy), by comparing predicted $\hat{v}_z$ to pseudo-ground-truth labels $v_z^*$ obtained using a GPT-4.1 oracle. The results shown in Table 5 suggest that VS maintains a strong advantage over the in-context learning baseline (Many2Many) under this natural error distribution.

Table 6: Answer accuracy on 1000 PRONTOQA problems. Standard deviation is reported on five intendant runs.

| Correction Strategy | Accuracy ↑ |
|---|---|
| No Correction (Raw CoT) | $0.712 \pm 0.002$ |
| Correction using Many2Many | $0.697 \pm 0.008$ |
| Correction using AVI | $\mathbf{0.730 \pm 0.002}$ |

Table 7: Reasoning accuracy on COMMONSENSEQA with one iteration of Self-Refine. Accuracy is averaged over 5 random seeds (± standard deviation).

| Method | Accuracy ↑ |
|---|---|
| Original Reasoning | $0.741 \pm 0.001$ |
| Self-Refine (Many2Many) | $0.749 \pm 0.005$ |
| Self-Refine (AVI) | $\mathbf{0.756 \pm 0.002}$ |

## C.2 SELF-CORRECTION/IMPROVEMENT OF LM-GENERATED REASONING CHAINS

Following from the experiment detailed in §C.1.2, where VS was applied to naturally-occurring errors, we then sought to evaluate whether AVI can be used as a feedback signal as part of a self-correction/improvement framework for improving reasoning performance. Concretely, we fine-tuned a Qwen3-4B model using AVI on predicted veracities sampled using VS for $10,000$ $(x, z, y^*)$ triples, to learn a distribution $Q(v_z \mid x, z)$ that does not depend on $y^*$. The training setup is otherwise identical to the one described in §2.3. Then, we used AVI to predict erroneous steps in $z$ by inferring stepwise veracities $\hat{v}_z \sim Q(v_z \mid x, z)$.

In the case of PRONTOQA, we corrected steps predicted to be `False` by negating them, and queried a Qwen3-4B model to predict the answer $y$ conditioned on the corrected CoT.

Table 6 shows that standard self-correction using in-context learning (Many2Many) to provide feedback via veracity assignments does not improve performance in PRONTOQA: the accuracy even drops from 0.712 to 0.697. This highlights that applying correction without reliable veracity identification may in fact be harmful. In contrast, our AVI-based correction increases accuracy to 0.730, showing that once veracity is accurately identified, correction becomes beneficial rather than detrimental. Without accurate veracity signals, correction may propagate or even amplify errors.

Next, we moved beyond simple correction via negation by testing the applicability of our approach in a more realistic setting where negation-based edits aren't a natural way to correct. In particular, we consider COMMONSENSEQA, where reasoning chains involve open-ended statements in natural language, and correction is more easily conducted as part of inference-time self-improvement methods such as *Self-Refine* (Madaan et al., 2023), which iteratively alternates between a *feedback model* and a *refinement model*. We adapt this framework by replacing the baseline feedback model (which is a few-shot prompted LM, similar to our Many2Many baseline) with our AVI machine. Given a CoT $z$ generated by Qwen3-4B, the AVI model first predicts veracity labels $v_z$ that identify incorrect reasoning steps. The refinement model (a few-shot-prompted Qwen3-4B model) then resamples downstream reasoning steps, conditioned on these veracity labels. This allows AVI to supply explicit supervision on *what to correct*, while the refinement model handles the problem of *how to correct*.

The results are shown in Table 7 for 1,000 test questions from COMMONSENSEQA. The gain from integrating AVI into Self-Refine is nearly twice as large as the improvement obtained with a standard Many2Many feedback model (+0.008), i.e. $0.015/0.008 \approx 1.9\times$, though in absolute terms may appear modest (+0.015 over the raw baseline). In general, the trend observed in all our experiments is that our method improves the accuracy of error-identification (whether using VS or AVI) by a larger margin over baselines than it improves reasoning accuracy post-correction, hinting at a bottleneck in reasoning capabilities in the underlying model. This reflects the fact that verification may be easier than generation, and that more work is needed to find ways to make better use of the more robust veracity signal provided by VS and AVI in downstream reasoning tasks.

## C.3 BLOCK METROPOLIS

Our default implementation of VS uses single-bit Metropolis updates, where one veracity label is flipped at a time. While simple and effective, this approach may be less effective if errors are correlated, requiring simultaneous updates to transition between distinct high-reward modes. To examine this possibility, we extended our sampler with *block Metropolis* updates, in which random contiguous blocks of size 1, 2, or 3 are flipped together.

Table 8: Average Hamming similarity of predicted veracity vectors using single-bit vs. block Metropolis updates on 1,000 samples with Qwen3-4B.

| Method | PRONTOQA | GSM8K | COMMONSENSEQA |
|---|---|---|---|
| Single-bit Metropolis | 0.910 | 0.711 | 0.860 |
| Blocked Metropolis | 0.918 | 0.704 | 0.867 |

As shown in Table 8, both variants perform comparably across the three tasks. This suggests that single-bit updates are already sufficient for short to medium-length reasoning chains in the domains of reasoning that we tested. Nonetheless, block updates represent a natural extension of our framework, with potential advantages for longer sequences and settings where LMs produce errors with more complex inter-dependencies.

Our approach is compatible with other proposal strategies, such as adaptive proposals, or gradient-informed updates, making them promising directions for future exploration.

## C.4 EXTENDING TO CATEGORICAL VARIABLES

In many contexts, binary veracity labels may be too restrictive. For example, when verifying properties such as the *relevance* of a reasoning step, or when allowing for ambiguity in correctness, it can be useful for the latent variable $V_z$ to take on more than two values. While this enlarges the label space, it remains far smaller than the full space of natural language sequences, making inference tractable.

Our framework naturally generalizes to this setting by replacing the binary vector with a $k$-class categorical vector and augmenting the MCMC transition table accordingly. To illustrate, we extended PRONTOQA with a third label, "Unrelated," by injecting unrelated—but factually correct—statements into the reasoning chain from a small predefined pool (e.g., "Humans are animals." or "The sky is blue.").

Table 9: Veracity prediction on PRONTOQA with three classes: True, False, and Unrelated. Average Hamming similarity across 100 samples is reported.

| Method | Hamming Similarity ↑ |
|---|---|
| Many2Many | 0.66 |
| VS | **0.91** |

These preliminary results suggest that our approach extends naturally beyond binary veracity, and can accommodate richer categorical variables for reasoning verification.

## C.5 CORRELATION ANALYSIS

Our working hypothesis (see Section 2) is that veracity assignments $v_z$ which maximize the joint-likelihood proxy reward

$$P_{\mathrm{LM}}(v_z \, y^* \mid x, z) \ \propto \ \mathbb{P}(V_z = v_z \mid Y = y^*, x, z)$$

will tend to be closer to the ground-truth veracity $v_z^*$. Intuitively, if the accuracy of a language model's final prediction depends on the correctness of its reasoning steps, then higher joint likelihood should correlate with more accurate veracity labels.

Table 10: Pearson correlation between Hamming similarity and joint likelihood across all possible veracity assignments (7 statements per sample). Results are averaged over 100 samples for each dataset.

| Model | PRONTOQA | COMMONSENSEQA |
|---|---|---|
| Qwen3-4B | 0.56 | 0.72 |
| Qwen3-8B | 0.74 | 0.78 |
| LLaMA 3B | 0.72 | 0.67 |
| LLaMA 8B | 0.70 | 0.79 |

Table 11: Average VS wall-time over 5 independent runs as a function of CoT length.

| Number of Reasoning Steps | Time (sec) |
|---|---|
| 3 | 7.33 |
| 5 | 7.83 |
| 7 | 11.14 |
| 9 | 12.19 |
| 11 | 13.09 |

To validate this hypothesis, we measured the correlation between the joint likelihood and veracity accuracy. For each problem we enumerated all possible veracity vectors ($2^N$ assignments for $N{=}7$ reasoning steps) and computed both their joint likelihood $P_{\text{LM}}(v_z\, y^* \mid x, z)$ and their Hamming similarity with the ground-truth vector $v_z^*$. Table 10 reports the average Pearson correlation across 100 randomly selected problems drawn from PRONTOQA and COMMONSENSEQA datasets.

These results suggest that veracity vectors with higher joint likelihood also achieve higher Hamming similarity with the ground truth. In other words, the proxy reward used in our search procedure is not only theoretically motivated but also empirically aligned with veracity accuracy. This correlation provides a clear post-hoc explanation for why our method consistently yielded strong Hamming similarity throughout the main experiments, thereby grounding the observed performance gains in a measurable property of language models.

## C.6 THE COST OF VERACITY SEARCH ON INFERENCE-TIME

In VS, the CoT $z$ is fixed, and we score a proposed veracity assignment $v_z$ via the joint likelihood $P_{\text{LM}}(v_z, y^* \mid x, z)$, where $y^*$ is the correct answer to the question in $x$. This likelihood is computed with teacher-forcing: the concatenated sequence $(v_z, y^*)$ is processed in a single forward pass (prefill stage), and logits for all tokens are produced under causal self-attention. This avoids the substantially more expensive sequential autoregressive decoding required by in-context learning baselines (Many2Many). However, because VS typically requires scoring multiple proposals as part of the search procedure, this advantage diminishes as the number of VS iterations increases. Empirically, we find that our method provides a favorable trade-off between inference time and veracity accuracy, as shown in Figure 4, but we provide a complementary analysis of the computational cost incurred from VS iterations below.

Let $L_{\text{context}} = |(x, z)|$ and $L_{\text{tail}} = |(v_z, y^*)|$. Naively, across $N$ VS iterations, the cost scales as $O\big(N(L_{\text{context}} + L_{\text{tail}})^2\big)$ due to the quadratic cost of self-attention. We can reduce computational requirements with prefix key-value caching for the fixed context $(x, z)$, so the overall cost becomes $O\big(L_{\text{context}}^2\big) + O\big(N(L_{\text{context}} L_{\text{tail}} + L_{\text{tail}}^2)\big)$. For proposals that differ from a reference $v_z$ at a single position $i$, we can also cache the extended prefix $(x, z, v_z[: i])$. If $i$ is uniformly distributed, the expected suffix length is $L_{\text{tail}}/2$, yielding a constant-factor reduction: $O\big(L_{\text{context}}^2\big) + O\big(N(L_{\text{context}} L_{\text{tail}}/2 + L_{\text{tail}}^2/4)\big)$. Finally, batching proposals that share the same cached prefix (or the same divergence index $i$) lets the GPU process many tails in parallel, improving utilization and reducing wall-clock time. In other words, we can trade memory for faster runtime, if desired.

We empirically evaluate how wall-clock time varies as a function of CoT length using Qwen3-4B-based VS in PRONTOQA, for CoTs with lengths ranging from 3 to 11 steps. A single GPU (RTX

A6000 ADA) was used, and VS involved 100 Metropolis iterations. The measurements displayed in Table 11 reflect end-to-end VS, including greedy initialization and simulated annealing.

## C.7 VERACITY SEARCH ON SMALLER AND LARGER MODELS

The original Veracity Search (VS) experiments were performed on 3B–8B models. To examine model-size sensitivity, we additionally evaluated a smaller model (Qwen3-1.7B) and a larger model (Qwen3-14B). The experimental design and search hyperparameters match those used to produce the results in Table 1.

Table 12: Veracity inference accuracy (Hamming similarity) across three benchmarks (average over 1000 samples each).

| | GSM8K | | ProntoQA | | CommonsenseQA | |
|---|---|---|---|---|---|---|
| | 1.7B | 14B | 1.7B | 14B | 1.7B | 14B |
| Many2Many | 0.512 | 0.675 | 0.529 | 0.714 | 0.488 | 0.524 |
| VS | **0.657** | **0.833** | **0.713** | **0.928** | **0.832** | **0.980** |

Across all three tasks, larger models provide more accurate veracity predictions when using either VS or the in-context learning baseline (Many2Many). VS maintains a consistent advantage over this baseline, indicating that the joint-likelihood used to guide VS becomes increasingly informative as model capacity increases.

## LARGE LANGUAGE MODEL USAGE

Large language models (LLM) were used only for minor polishing of the writing quality both in the main text and in the code. They were also used to assist with debugging. Any modification to the text that was suggested by an LLM to improve clarity was verified by the authors. LLMs were not used to generate the ideas, methods, experimental designs, and analyses of results that are presented in this paper.

