# OpenReview forum: "Latent Veracity Inference for Identifying Errors in Stepwise Reasoning"
_ICLR.cc/2026/Conference — ICLR 2026 Poster_

### Official Review · Reviewer_xAHw · 2025-10-29

**Soundness:** 3
**Presentation:** 3
**Contribution:** 3
**Rating:** 6
**Confidence:** 2

**Summary:**

This paper focuses on identifying stepwise error in Chain of Thoughts (CoT) of language models. To address this problem, the paper formulates the problem as a latent variable model, and proposes two methods: veracity search (VS) and Amortized Veracity Inference (AVI) to solve the problem.

**Strengths:**

1. The proposed method provides significant improvement over the stepwise error detection performance.
2. The inference time cost of the proposed AVI method is relatively low.

**Weaknesses:**

1. The experiments are done with synthetic errors. I'm curious about how the proposed methods work for "real" errors in model's CoT. Can the AVI method detect and correct real CoT errors thus improve the reasoning ability?

**Questions:**

1. Can the empirical study in appendix C.6 be extended to other models/datasets? I think the hypothesis here is actually key reason why the proposed methods work: the model actually "knows" the error and assign higher probability to the joint distribution of answer and veracity, if it's closer to ground truth. Providing more empirical evidence on this will strongthen the conclusion.

---

> ### Author Response · Authors · 2025-11-21
>
> ### **A1: The experiments are done with synthetic errors. I'm curious about how the proposed methods work for "real" errors in model's CoT. Can the AVI method detect and correct real CoT errors thus improve the reasoning ability?**
>
> We agree that evaluating on naturally generated CoTs is essential for understanding real-world behavior. To address this, we evaluated VS performance in natural settings for both ProntoQA and CommonsenseQA. In these evaluations, CoTs are generated by a Qwen3-4B model with no synthetic corruption; all reasoning errors therefore arise naturally from the model’s own limitations.
>
> We treat each sentence as an independent proposition and append a simple {T/F} veracity tag to each of them. Step‑level pseudo‑ground‑truth veracity labels are obtained from a GPT‑4.1 oracle. Veracity accuracy for our Veracity Search (VS) method and the few‑shot prompting baseline (Many2Many) is shown below:
>
>
> **Veracity Accuracy on Naturally Generated CoTs in ProntoQA (1,000 samples)**
>
> | Method    | Hamming Sim. | Exact Match |
> | --------- | ------------ | ----------- |
> | Many2Many |      0.74    |  0.00    |
> | VS        |     0.92   |  0.50    |
>
> **Veracity Accuracy on Naturally Generated CoTs in CommonsenseQA (1,000 samples)**
>
> | Method    | Hamming Sim. | Exact Match |
> | --------- | ------------ | ----------- |
> | Many2Many | 0.59         | 0.00        |
> | VS        | 0.86         | 0.59        |
>
> ---
>
> We further assess whether latent‑veracity inference improves downstream reasoning on natural CoTs generated by Qwen3‑4B. In ProntoQA, we used a verification model trained via Amortized Veracity Inference (AVI) to identify false propositions in a reasoning trace and corrected them by negation, whereas for CommonsenseQA, we adopt **a more general self‑improvement approach**, specifically the **Self‑Refine** method. We use the AVI-trained model (or the Many2Many baseline) to provide veracity predictions, and use this as feedback for the refinement model (also Qwen3-4B) to regenerate all steps after a predicted reasoning error:
>
> **ProntoQA Reasoning Accracy (Negation, 1 iteration, 1,000 samples)**
> *(mean ± std across 5 seeds)*
>
> | Method                  | Accuracy      |
> | ----------------------- | ------------- |
> | Original Reasoning      | 0.712 ± 0.002 |
> | Negation (Many2Many) | 0.697 ± 0.008 |
> | Negation (AVI)       | 0.730 ± 0.002 |
>
>
>
> **CommonsenseQA Reasoning Accuracy (Self-Refine, 1 iteration, 1,000 samples)**
> *(mean ± std across 5 seeds)*
>
> | Method                  | Accuracy      |
> | ----------------------- | ------------- |
> | Original Reasoning      | 0.741 ± 0.001 |
> | Self-Refine (Many2Many) | 0.749 ± 0.005 |
> | Self-Refine (AVI)       | 0.756 ± 0.002 |
>
> Together, these results extend those in our original submission and provide further evidence that our proposed method generalizes to naturally occurring reasoning errors.

---

> ### Author Response · Authors · 2025-11-21
>
> ### **Q1: Can the empirical study in appendix C.6 be extended to other models/datasets? I think the hypothesis here is actually key reason why the proposed methods work: the model actually "knows" the error and assign higher probability to the joint distribution of answer and veracity, if it's closer to ground truth. Providing more empirical evidence on this will strongthen the conclusion.**
>
> We appreciate the suggestion to broaden our correlation analysis. Below, we extend the study by measuring the correlation between the joint likelihood used by VS, $P_{\mathrm{LM}}(v_z, y^\star \mid x, z)$, and the Hamming similarity between $v_z$ and the (pseudo‑)ground truth $v_z^\star$. Ground‑truth labels are used when available (only for ProntoQA); otherwise, we use pseudo‑labels generated by a GPT‑4.1 oracle:
>
>
> **Expanded correlation results**
>
>
> |  | ProntoQA | CommonsenseQA |
> | -------- | -------- | -------- |
> | Qwen 4B     | 0.56   | 0.72    |
> | Qwen 8B     | 0.74    | 0.78    |
> | llama 3B     |0.72    | 0.67     |
> | Llama 8B     | 0.70    | 0.79    |
>
> We will include the expanded results in the revised Appendix C.6.

---

### Official Review · Reviewer_zvRp · 2025-10-31

**Soundness:** 3
**Presentation:** 3
**Contribution:** 2
**Rating:** 6
**Confidence:** 4

**Summary:**

This paper presents an approach to disentangle the content of a chain-of-thought (CoT) from its veracity/correctness. The authors introduce veracity search (VS), which is a metropolis search methodthat samples veracity assignments for reasoning steps using the LM’s joint likelihood as a reward, and amortized veracity inference (AVI), which fine-tunes an LM on pseudo-labels from VS to enable zero-shot veracity prediction. The idea of treating correctness as a latent variable is both intuitive and novel, and the experiments convincingly demonstrate gains over prompting-based verification baselines. The paper is well written and includes extensive experiments across logical, math, and commonsense reasoning.

**Strengths:**

1. **Conceptual clarity and originality**: The paper disentangles reasoning content and correctness via a latent variable formulation. It provides a neat probabilistic framing of step-wise error identification in chain of thoughts.
2. **veracity search is intuitive**: and effective inference-time algorithm. It  outperforms simple prompting-based verifiers across different benchmarks.
3. **Comprehensive evaluation.** The paper is well written, and the experiments are well-organized, and includes detailed ablations (e.g., simulated annealing, greedy initialization, scalability with reasoning hops on prontoQA)..

**Weaknesses:**

1. **Reliance on prompting LMs for veracity scoring:** The entire framework assumes that the LM can reliably evaluate joint likelihoods of veracity assignments, yet prior studies (e.g., Huang et al., 2023; Zhang et al., 2024) show that LMs are often poor self-verifiers, especially on real-world reasoning where correctness is tricky/subtle. Some discussion or empirical evidence of robustness on naturally occurring reasoning errors would strengthen the claims.

2. **Lack of comparison to process reward models (PRMs).** There exist strong baselines such as PRMs (Lightman et al., 2024; Cobbe et al., 2021) that explicitly model step-level correctness and can perform veracity inference without search. This begs the question of why should VS/AVI be preferred over training or using a PRM?
PRMs would likely achieve comparable or better results with far fewer LM forward passes and a clearer training signal.

3. The search process requires up to 200 LM forward passes per example, making it orders of magnitude more expensive than single-pass prompting baselines so the comparison to these is unfair in my opinion. While AVI mitigates this at inference, the paper should discuss the practical feasibility and compute trade-offs, especially when compared to PRMs or trained verifiers.

4. **Limited practical applicability:** Many of the ablations/analysis focus on synthetic or structured boolean reasoning over prontoQA. It remains unclear whether the method can handle open-ended or natural reasoning errors (e.g., in mathematical or real-world proofs). The negation-based correction strategy also seems tied to synthetic logic tasks.

**Questions:**

See above

---

> ### Author Response · Authors · 2025-11-21
>
> ### **W1: Robustness on naturally occurring reasoning errors and relation to prior self-verification studies**
>
> We appreciate the concern that LLMs can be poor self‑verifiers in real‑world reasoning tasks. Prior work (e.g., Huang et al., ICLR ’24) shows that in “intrinsic” self‑correction (without oracle labels to decide when a correction is needed), few‑shot prompted critique/verification models can be fragile and even degrade performance. This is a known limitation of turning LLMs into self‑verifiers via few‑shot prompting (our Many2Many baseline).
>
> Our method differs fundamentally. We do not directly prompt the LM to judge the veracity of its own steps. The LM provides only a joint‑likelihood $P_{\mathrm{LM}}(v_z, y^\star \mid x, z)$ over step‑level veracity assignments $v_z$ and the ground‑truth final answer $y^\star$, while veracity inference is performed externally by Veracity Search (VS), which approximates the target latent‑variable posterior. In short, we use the LM’s joint likelihood to extract a statement‑level signal, but the joint veracity assignment is determined by a search procedure targeting the intractable distribution of interest, rather than by sampling from a tractable, few‑shot‑prompted surrogate that empirically appears to be a poor approximation. In addition, VS requires only minimal prompting, e.g.,
>
> ```
> You are a logical reasoner.
> Label True if a step follows from Context, False otherwise.
> ```
>
> reducing the potential for prompt sensitivity. This contrasts with many in‑context learning methods, including open‑ended critique/debate approaches, which Huang et al. identify as prompt‑sensitive and less reliable for verification.
>
> Our work also align with the other prior work cited by the reviewer (SCORE; Zhang et al., 2025). Similar to SCORE, which decomposes self‑correction into VERIFY and REFINE components, we split the task into a veracity‑inference step (via VS or our Amortized Veracity Inference, AVI) and a refinement step (via Self‑Refine). Of these two modules, Zhang et al. find that verification is typically the bottleneck, with verifier accuracy tightly tracking final‑answer accuracy.
>
> In keeping with their findings, we observe that the VS proxy reward correlates more strongly with veracity accuracy in larger (more capable) models compared to smaller ones (Appendix C.6).
>
> We will expanded the introduction and related‑work sections to include the above discussion.
>
> Furthermore, we also provide an empirical analysis of naturally occurring errors; please see our response to W4 in the final section below.
>
> [1] Huang et al., "Large Language Models Cannot Self-Correct Reasoning Yet", ICLR 2024
> [2] Zhang et al., "Small Language Models Need Strong Verifiers to Self-Correct Reasoning", ACL 2024

---

> > ### Author Response · Authors · 2025-11-21
> >
> > ### **W2: Lack of comparison to process reward models (PRMs)**
> >
> > We thank the reviewer for raising this point. Process Reward Models (PRMs), which numerically score the quality of intermediate reasoning steps and have become an important part of LM post‑training. Our Amortized Veracity Inference (AVI) method also trains an LM to output a per‑step signal (in our case, a probability of veracity), and so it is quite reasonable to want to compare the two. However, two key differences make an apples‑to‑apples comparison difficult:
> >
> > 1. Traditional PRM approaches, including the one cited by the reviewer (Lightman et al., 2023), require process supervision: labeled step‑level rewards indicating the quality or correctness of each intermediate step. These labels are costly to obtain and highly domain‑specific (math, logic, coding each require distinct annotation schemes). By contrast, our methods (VS and AVI) do not rely on step‑level supervision; we treat step‑level veracity as latent and use the LM’s joint likelihood over latent veracity assignments and the final answer to infer veracity labels consistent with the latent‑variable model in Section 2.1. The other paper cited by the reviewer (Cobbe et al., 2021) is arguably better characterized as an Outcome Reward Model (ORM), since it produces a global score for the entire solution rather than per‑step rewards, and is therefore not directly comparable to AVI.
> >
> > 2. There are also PRM approaches that derive “implicit” stepwise rewards from outcome supervision alone (final‑answer labels), trained via SFT or RL and sometimes interleaved with updates to the generative model that produces the reasoning traces [1–4]. These unsupervized PRMs capture the usefulness of reasoning steps in arriving at the final answer (such as the expected return from a step), rather than the truth of that step, which is quite different in principle. Such a PRM could assign high reward to steps that tend to precede correct answers, even if those steps are not themselves correct. For example, reasoning steps that explore false premises but then later retract them, can still be useful despite being incorrect.
> >
> > To summarize: PRMs trained with reasoning correctness annotations (point 1) can be viewed as approximate veracity predictors, while outcome derived PRMs (point 2) somewhat conflate veracity and instrumental value. Despite the lack of supervision and a simliar source of signal, our methods (VS/AVI) are positioned to make this distinction because they exploit a more subtle training signal: not hypothezied correlation between the correctness of reasoning steps and correct answers, but rather hypothesized correlation between correct answers and a LM's ability to assess that a step it has just taken is correct.
> >
> > As a result of the reviewer's question, we have been asking ourselves whether our methods (VS/AVI) could form the basis of a new PRM method. But doing this properly would be a significant extention and remains, for now, an avenue for future investigation.
> >
> >
> > [1] Uesato J, et al. Solving math word problems with process- and outcome-based feedback. 2022.
> > [2] Yuan L, et al. Free Process Rewards Without Process Labels. ICML 2025.
> > [3] Cui G, et al. Process Reinforcement Through Implicit Rewards. 2025.
> > [4] Zha K, et al. RL Tango: Reinforcing Generator and Verifier Together for Language Reasoning. NeurIPS 2025.

---

> ### Author Response · Authors · 2025-11-21
>
> ### **W3: The search process requires up to 200 LM forward passes per example, making it orders of magnitude more expensive than single-pass prompting baselines so the comparison to these is unfair in my opinion. While AVI mitigates this at inference, the paper should discuss the practical feasibility and compute trade-offs, especially when compared to PRMs or trained verifiers.**
>
> Thank you for raising this important point. For a fair comparison, it is crucial to align compute budgets across methods. In our inference‑time analysis (Figure 4), we matched resources by allowing the few‑shot prompting baseline (Many2Many) to use majority voting over multiple samples. Concretely, when VS is run for 100 iterations, we compared against a Many2Many variant that draws 100 samples from its conditional distribution and uses majority voting over the sampled veracity assignments. This experiment is shown in Figure 4a (the x‑axis is the number of iterations/samples for all methods). Under matched sampling budgets, VS achieves higher veracity accuracy. We hope this clarifies the fairness of our comparison, and we welcome alternative suggestions.
>
> It is also noteworthy that VS achieves higher performance without a significant runtime penalty (Figure 4b). To clarify why, we analyze the computational complexity of VS in more detail. In VS, the CoT $z$ is fixed, and we score a proposed veracity assignment $v_z$ via the joint likelihood $P_{\mathrm{LM}}(v_z, y^\star \mid x, z)$, where $y^\star$ is the correct answer to the question in $x$. This likelihood is computed with teacher‑forcing: the concatenated sequence $(v_z, y^\star)$ is processed in a single forward pass (prefill), and logits for all tokens are produced under causal self‑attention. This avoids the (substantially more expensive) sequential autoregressive decoding required by in‑context learning baselines (Many2Many). That said, because VS typically requires scoring multiple proposals as part of the search, this advantage diminishes as the number of VS iterations increases. Empirically, we find a favorable trade‑off between inference time and veracity accuracy (Figure 4).
>
>
> To describe the computational complexity in more detail, let $L_\text{context} = |(x, z)|$ and $L_\text{tail} = |(v_z, y^\star)|$. Naively, across $N$ VS iterations, the cost scales as $O\big(N (L_\text{context} + L_\text{tail})^2\big)$. We can reduce compute with prefix key‑value caching by computing the fixed $(x, z)$ once and reusing it, so the overall cost becomes $O\big(L_\text{context}^2\big) + O\big(N (L_\text{context}\, L_\text{tail} + L_\text{tail}^2)\big)$. For proposals that differ from a reference $v_z$ at a single position $i$, we can also cache the extended prefix $(x, z, v_z[:i])$. If $i$ is approximately uniform, the expected suffix length is $L_\text{tail}/2$, yielding a constant‑factor reduction: $O\big(L_\text{context}^2\big) + O\big(N (L_\text{context}\, L_\text{tail}/2 + L_\text{tail}^2/4)\big)$. Finally, batching proposals that share the same cached prefix (or the same divergence index $i$) lets the GPU process many tails in parallel, improving utilization and reducing wall‑clock time. I.e., we can trade memory for faster runtime, if desired.
>
> Regarding AVI vs. PRMs, please see our response above. We agree that a comparison would be valuable if per‑step PRM rewards corresponded to per‑step correctness; however, to our knowledge, only process‑supervised PRMs enable such veracity inference, and our experimental setting assumes no access to stepwise veracity labels.

---

> ### Author Response · Authors · 2025-11-21
>
> ### **W4: Limited practical applicability: Many of the ablations/analysis focus on synthetic or structured boolean reasoning over prontoQA. It remains unclear whether the method can handle open-ended or natural reasoning errors (e.g., in mathematical or real-world proofs). The negation-based correction strategy also seems tied to synthetic logic tasks.**
>
> We agree that evaluating on naturally generated CoTs is essential for understanding real-world reasoning behavior. To address this, we first evaluate VS on **natural (non-synthetic)** CoTs for ProntoQA, replacing the synthetic perturbations used in our original experiments. To further alleviate concerns that our analysis relied primarily on a logical dataset, we additionally evaluate VS on **CommonsenseQA**, a non-formal, open-ended reasoning task.
>
> In both settings, CoTs are produced by a Qwen3-4B model **without any synthetic corruption**, so all reasoning errors arise naturally from the model’s own limitations. For evaluation, we obtain step-level pseudo-ground-truth veracity labels are obtained from a GPT-4.1 oracle. Veracity accuracy for our Veracity Search (VS) method and the few-shot prompting baseline (Many2Many) is shown below:
>
> **Veracity Accuracy on Naturally Generated CoTs in ProntoQA (1,000 samples)**
>
> | Method    | Hamming Sim. | Exact Match |
> | --------- | ------------ | ----------- |
> | Many2Many |      0.74    |  0.00    |
> | VS        |     0.92   |  0.50    |
>
> **Veracity Accuracy on Naturally Generated CoTs in CommonsenseQA (1,000 samples)**
>
> | Method    | Hamming Sim. | Exact Match |
> | --------- | ------------ | ----------- |
> | Many2Many | 0.59         | 0.00        |
> | VS        | 0.86         | 0.59        |
>
> ---
>
> We further assess whether latent-veracity inference improves downstream reasoning on natural CoTs generated by Qwen3-4B. For ProntoQA, we use a verification model trained via Amortized Veracity Inference (AVI) to identify false propositions in the reasoning trace and correct them via simple negation. For CommonsenseQA, where negation is not applicable, we adopt a more general self-improvement mechanism, specifically the Self-Refine method.
>
> Self-Refine is not restricted to formal logical tasks and naturally extends to open-ended reasoning: the AVI-trained verifier (or the Many2Many baseline) provides veracity predictions, which are then used as feedback by the refinement model (also Qwen3-4B) to regenerate all steps after the first predicted reasoning error:
>
> **ProntoQA Reasoning Accracy (Negation, 1 iteration, 1,000 samples)**
> *(mean ± std across 5 seeds)*
>
> | Method                  | Accuracy      |
> | ----------------------- | ------------- |
> | Original Reasoning      | 0.712 ± 0.002 |
> | Negation (Many2Many) | 0.697 ± 0.008 |
> | Negation (AVI)       | 0.730 ± 0.002 |
>
>
>
> **CommonsenseQA Reasoning Accuracy (Self-Refine, 1 iteration, 1,000 samples)**
> *(mean ± std across 5 seeds)*
>
> | Method                  | Accuracy      |
> | ----------------------- | ------------- |
> | Original Reasoning      | 0.741 ± 0.001 |
> | Self-Refine (Many2Many) | 0.749 ± 0.005 |
> | Self-Refine (AVI)       | 0.756 ± 0.002 |
>
> Together, these results extend those in our original submission and provide further evidence that our proposed method generalizes to naturally occurring reasoning errors.

---

### Official Review · Reviewer_Qqpf · 2025-10-31

**Soundness:** 3
**Presentation:** 3
**Contribution:** 3
**Rating:** 6
**Confidence:** 3

**Summary:**

This paper proposed augmenting each CoT step with a latent correctness variable and a search method, Veracity Search to infers step correctness by maximizing the language model’s joint likelihood of veracity and the final answer

**Strengths:**

1. strong performance gain
2. label-efficient step verification, the proposed veracity search use LLM's joint likelihood, avoiding expensive step-level supervision.

**Weaknesses:**

1. Most tests use artificially corrupted chains; evidence on naturally occurring errors is limited and needs broader experiments.
2. The joint-likelihood reward correlates with true veracity but not perfectly (Pearson 0.56–0.74), so misrankings can occur.
3. the computation efficiency can be improved, strong VS performance often use tens to 100 samples, this add-on computation cost compared with single-pass verifiers.

**Questions:**

N/A

---

> ### Author Response · Authors · 2025-11-21
>
> ### **W1: Most tests use artificially corrupted chains; evidence on naturally occurring errors is limited and needs broader experiments.**
>
> We agree that evaluating on naturally generated CoTs is essential for understanding real-world behavior. To address this, we evaluated VS performance in natural settings for both ProntoQA and CommonsenseQA. In these evaluations, CoTs are generated by a Qwen3-4B model with no synthetic corruption; all reasoning errors therefore arise naturally from the model’s own limitations.
>
> For evaluation, we obtain step-level pseudo-ground-truth veracity labels are obtained from a GPT-4.1 oracle. Veracity accuracy for our Veracity Search (VS) method and the few-shot prompting baseline (Many2Many) is shown below:
>
> **Veracity Accuracy on Naturally Generated CoTs in ProntoQA (1,000 samples)**
>
> | Method    | Hamming Sim. | Exact Match |
> | --------- | ------------ | ----------- |
> | Many2Many |      0.74    |  0.00    |
> | VS        |     0.92   |  0.50    |
>
> **Veracity Accuracy on Naturally Generated CoTs in CommonsenseQA (1,000 samples)**
>
> | Method    | Hamming Sim. | Exact Match |
> | --------- | ------------ | ----------- |
> | Many2Many | 0.59         | 0.00        |
> | VS        | 0.86         | 0.59        |
>
> We further assess whether latent‑veracity inference improves downstream reasoning on natural CoTs generated by Qwen3‑4B. In ProntoQA, we used a verification model trained via Amortized Veracity Inference (AVI) to identify false propositions in a reasoning trace and corrected them by negation, whereas for CommonsenseQA, we adopt **a more general self‑improvement approach**, specifically the **Self‑Refine** method. We use the AVI-trained model (or the Many2Many baseline) to provide veracity predictions, and use this as feedback for the refinement model (also Qwen3-4B) to regenerate all steps after a predicted reasoning error:
>
> **ProntoQA Reasoning Accracy (Negation, 1 iteration, 1,000 samples)**
> *(mean ± std across 5 seeds)*
>
> | Method                  | Accuracy      |
> | ----------------------- | ------------- |
> | Original Reasoning      | 0.712 ± 0.002 |
> | Negation (Many2Many) | 0.697 ± 0.008 |
> | Negation (AVI)       | 0.730 ± 0.002 |
>
>
>
> **CommonsenseQA Reasoning Accuracy (Self-Refine, 1 iteration, 1,000 samples)**
> *(mean ± std across 5 seeds)*
>
> | Method                  | Accuracy      |
> | ----------------------- | ------------- |
> | Original Reasoning      | 0.741 ± 0.001 |
> | Self-Refine (Many2Many) | 0.749 ± 0.005 |
> | Self-Refine (AVI)       | 0.756 ± 0.002 |
>
> Together, these results extend those in our original submission and provide further evidence that our proposed method generalizes to naturally occurring reasoning errors.

---

> ### Author Response · Authors · 2025-11-21
>
> ### **W2: The joint-likelihood reward correlates with true veracity but not perfectly (Pearson 0.56–0.74), so misrankings can occur.**
>
> We agree that the correlation between the joint‑likelihood reward and true veracity is not perfect, and that mis-rankings can occur. More broadly, this reflects a key limitation of self‑verification/self‑improvement methods: gains are upper‑bounded by the model’s verification capabilities. This applies to both our Veracity Search (VS) method and the baselines we consider. That said, the observed Pearson correlations (0.56–0.74) indicate a moderate‑to‑strong positive association between our proxy reward and veracity accuracy, and we find that larger models exhibit higher correlation, suggesting that as LMs improve, joint likelihood becomes increasingly informative for self‑verification. This trend is consistent with prior work on self‑verification (e.g., Zhang et al., 2024 [1]).
>
> To further analyze this trend, we expanded our correlation experiments to include LLaMA models and CommonsenseQA. As shown below, the 0.56 correlation for Qwen-4B on ProntoQA is an isolated low case; most settings yield correlations above 0.7, with all models ≥8B consistently exceeding 0.7.
>
> **Expanded correlation results (Pearson ↑ / Hamming similarity ↑)**
>
> | Model       | ProntoQA        | CommonsenseQA    |
> |-------------|------------------|------------------|
> | Qwen 4B     | 0.56 / **0.91** | 0.72 / **0.94** |
> | Qwen 8B     | 0.74 / **0.95** | 0.78 / **0.93** |
> | LLaMA 3B    | 0.72 / **0.95** | 0.67 / **0.84** |
> | LLaMA 8B    | 0.70 / **0.96** | 0.79 / **0.90** |
>
> Importantly, the observed correlation is already empirically sufficient for accurate inference: VS achieves consistently high Hamming similarity across tasks (typically above 0.9), indicating that the joint-likelihood reward reliably guides the search toward correct veracity assignments in practice. Further improving the proxy reward—e.g., by combining likelihood with auxiliary consistency signals—remains a promising direction for future work.
>
> [1] Zhang et al., "Small Language Models Need Strong Verifiers to Self-Correct Reasoning", ACL 2024
>
> ---
>
> ### W3: the computation efficiency can be improved, strong VS performance often use tens to 100 samples, this add-on computation cost compared with single-pass verifiers.
>
> We thank the reviewer for highlighting the importance of analyzing the impact of our method on inference time. In VS, the CoT $z$ is fixed, and we score a proposed veracity assignment $v_z$ via the joint likelihood $P_{\mathrm{LM}}(v_z, y^\star \mid x, z)$, where $y^\star$ is the correct answer to the question in $x$. This likelihood is computed with teacher‑forcing: the concatenated sequence $(v_z, y^\star)$ is processed in a single forward pass (prefill), and logits for all tokens are produced under causal self‑attention. This avoids the (substantially more expensive) sequential autoregressive decoding required by in‑context learning baselines (Many2Many). That said, because VS typically requires scoring multiple proposals as part of the search, this advantage diminishes as the number of VS iterations increases. Empirically, we find that our method provides a favorable trade‑off between inference time and veracity accuracy, as shown in Figure 4.
>
> To describe the computational complexity in more detail, let $L_\text{context} = |(x, z)|$ and $L_\text{tail} = |(v_z, y^\star)|$. Naively, across $N$ VS iterations, the cost scales as $O\big(N (L_\text{context} + L_\text{tail})^2\big)$. We can reduce compute with prefix key‑value caching by computing the fixed $(x, z)$ once and reusing it, so the overall cost becomes $O\big(L_\text{context}^2\big) + O\big(N (L_\text{context}\, L_\text{tail} + L_\text{tail}^2)\big)$. For proposals that differ from a reference $v_z$ at a single position $i$, we can also cache the extended prefix $(x, z, v_z[:i])$. If $i$ is approximately uniform, the expected suffix length is $L_\text{tail}/2$, yielding a constant‑factor reduction: $O\big(L_\text{context}^2\big) + O\big(N (L_\text{context}\, L_\text{tail}/2 + L_\text{tail}^2/4)\big)$. Finally, batching proposals that share the same cached prefix (or the same divergence index $i$) lets the GPU process many tails in parallel, improving utilization and reducing wall‑clock time. I.e., we can trade memory for faster runtime, if desired.
>
> Moreover, our Amortized Veracity Inference (AVI) further mitigates VS’s iterative cost by amortizing VS into a single‑pass verifier, yielding accuracy close to VS while reducing inference to a single LM forward pass.

---

### Official Review · Reviewer_knXB · 2025-11-01

**Soundness:** 3
**Presentation:** 3
**Contribution:** 3
**Rating:** 4
**Confidence:** 3

**Summary:**

This paper addresses the problem that chain-of-thought (CoT) reasoning in LLMs often contains flawed intermediate steps, which undermines both accuracy and interpretability. The authors propose to augment each reasoning step with a latent veracity variable indicating correctness. They introduce Veracity Search (VS), a discrete search algorithm that leverages the joint likelihood of veracity assignments and final answers to approximate posterior inference. To avoid reliance on ground-truth answers at test time, they further propose Amortized Veracity Inference (AVI), which is trained on pseudo-labels generated by VS and enables zero-shot veracity prediction.

**Strengths:**

(1) Originality: The methods proposed for improving the CoT are innovative.

(2) Quality: The combination of VS and AVI is well-motivated, and experiments are carefully designed across logical, mathematical, and commonsense reasoning tasks.

(3) Clarity: The paper provides clear definitions, and comprehensive experiments.

(4) Significance: Identifying and correcting reasoning errors is an important challenge for improving the reliability of LMs, and this work provides a promising direction.

**Weaknesses:**

(1) Many experiments rely on artificially corrupted reasoning chains. It would be valuable to see more extensive evaluation on naturally generated CoTs (this is the real-world use case).

(2) The research on the impact of reasoning time is somewhat lacking.

(3) The AVI is dependent on VS, but VS itself may not able to guarantee accuracy. This influence is not analyzed.

**Questions:**

(1) How well does the method generalize to naturally occurring errors in CoTs, beyond the controlled corruption schemes?

(2) Could the authors provide more discussion on the computational cost compared to baselines, especially for longer reasoning chains? The influence on reasoning time is not known.

(3) How do you think your proposed method will perform on larger or smaller LLMs? Your experiment only involves LLMs of approximately 4B or 8B.

---

> ### Author Response · Authors · 2025-11-21
>
> ### **W1 & Q1: Reliance on Artificial Corruption**
> > Many experiments rely on artificially corrupted reasoning chains. It would be valuable to see more extensive evaluation on naturally generated CoTs (this is the real-world use case).
>
> We agree that evaluating on naturally generated CoTs is essential for understanding real-world behavior. To address this, we evaluated VS performance in natural settings for both ProntoQA and CommonsenseQA. In these evaluations, CoTs are generated by a Qwen3-4B model with no synthetic corruption; all reasoning errors therefore arise naturally from the model’s own limitations.
>
> For evaluation, we obtain step-level pseudo-ground-truth veracity labels are obtained from a GPT-4.1 oracle. Veracity accuracy for our Veracity Search (VS) method and the few-shot prompting baseline (Many2Many) is shown below:
>
> **Veracity Accuracy on Naturally Generated CoTs in ProntoQA (1,000 samples)**
>
> | Method    | Hamming Sim. | Exact Match |
> | --------- | ------------ | ----------- |
> | Many2Many |      0.74    |  0.00    |
> | VS        |     0.92   |  0.50    |
>
> **Veracity Accuracy on Naturally Generated CoTs in CommonsenseQA (1,000 samples)**
>
> | Method    | Hamming Sim. | Exact Match |
> | --------- | ------------ | ----------- |
> | Many2Many | 0.59         | 0.00        |
> | VS        | 0.86         | 0.59        |
>
> ---
>
> We further assess whether latent‑veracity inference improves downstream reasoning on natural CoTs generated by Qwen3‑4B. In ProntoQA, we used a verification model trained via Amortized Veracity Inference (AVI) to identify false propositions in a reasoning trace and corrected them by negation, whereas for CommonsenseQA, we adopt **a more general self‑improvement approach**, specifically the **Self‑Refine** method. We use the AVI-trained model (or the Many2Many baseline) to provide veracity predictions, and use this as feedback for the refinement model (also Qwen3-4B) to regenerate all steps after a predicted reasoning error:
>
> **ProntoQA Reasoning Accracy (Negation, 1 iteration, 1,000 samples)**
> *(mean ± std across 5 seeds)*
>
> | Method                  | Accuracy      |
> | ----------------------- | ------------- |
> | Original Reasoning      | 0.712 ± 0.002 |
> | Negation (Many2Many) | 0.697 ± 0.008 |
> | Negation (AVI)       | 0.730 ± 0.002 |
>
>
>
> **CommonsenseQA Reasoning Accuracy (Self-Refine, 1 iteration, 1,000 samples)**
> *(mean ± std across 5 seeds)*
>
> | Method                  | Accuracy      |
> | ----------------------- | ------------- |
> | Original Reasoning      | 0.741 ± 0.001 |
> | Self-Refine (Many2Many) | 0.749 ± 0.005 |
> | Self-Refine (AVI)       | 0.756 ± 0.002 |
>
> Together, these results extend those in our original submission and provide further evidence that our proposed method generalizes to naturally occurring reasoning errors.

---

> ### Author Response · Authors · 2025-11-21
>
> ### **W2 & Q2: The Impact on Reasoning Time**
> > The research on the impact of reasoning time is somewhat lacking
>
> We thank the reviewer for highlighting the importance of analyzing the impact of our method on inference time. In VS, the CoT $z$ is fixed, and we score a proposed veracity assignment $v_z$ via the joint likelihood $P_{\mathrm{LM}}(v_z, y^\star \mid x, z)$, where $y^\star$ is the correct answer to the question in $x$. This likelihood is computed with teacher‑forcing: the concatenated sequence $(v_z, y^\star)$ is processed in a single forward pass (prefill), and logits for all tokens are produced under causal self‑attention. This avoids the (substantially more expensive) sequential autoregressive decoding required by in‑context learning baselines (Many2Many). That said, because VS typically requires scoring multiple proposals as part of the search, this advantage diminishes as the number of VS iterations increases. Empirically, we find that our method provides a favorable trade‑off between inference time and veracity accuracy, as shown in Figure 4.
>
> To describe the computational complexity in more detail, let $L_\text{context} = |(x, z)|$ and $L_\text{tail} = |(v_z, y^\star)|$. Naively, across $N$ VS iterations, the cost scales as $O\big(N (L_\text{context} + L_\text{tail})^2\big)$. We can reduce compute with prefix key‑value caching by computing the fixed $(x, z)$ once and reusing it, so the overall cost becomes $O\big(L_\text{context}^2\big) + O\big(N (L_\text{context}\, L_\text{tail} + L_\text{tail}^2)\big)$. For proposals that differ from a reference $v_z$ at a single position $i$, we can also cache the extended prefix $(x, z, v_z[:i])$. If $i$ is approximately uniform, the expected suffix length is $L_\text{tail}/2$, yielding a constant‑factor reduction: $O\big(L_\text{context}^2\big) + O\big(N (L_\text{context}\, L_\text{tail}/2 + L_\text{tail}^2/4)\big)$. Finally, batching proposals that share the same cached prefix (or the same divergence index $i$) lets the GPU process many tails in parallel, improving utilization and reducing wall‑clock time. I.e., we can trade memory for faster runtime, if desired.
>
> We evaluate wall‑clock time empirically using Qwen3‑4B on ProntoQA for CoTs with lengths ranging from 3 to 11 steps. VS uses 100 Metropolis iterations. The measurements below reflect end‑to‑end VS, including greedy initialization and simulated annealing:
>
>
> **Computational time average over 5 indepandant runs. Single GPU of RTX A6000 ADA is used**
>
> | Reasoning Chain Length| Time (100 iter) |
> | -------- | -------- |
> | 3  | 7.33s|
> | 5    | 7.83s     |
> | 7    | 11.14s     |
> | 9  |   12.19s  |
> | 11   |   13.09s  |
>
> We now include this more detailed analysis in the Appendix.
>
> ---
>
> ### **W3: The AVI is dependent on VS, but VS itself may not able to guarantee accuracy. This influence is not analyzed.**
>
> We agree that AVI’s performance depends on the quality of the pseudo‑labels produced by VS. More broadly, this reflects a key limitation of self‑verification/self‑improvement methods: gains are upper‑bounded by the model’s verification capabilities. This applies to VS and to the baselines we consider. For VS, we provide a correlation analysis in Appendix C.6, along with the **expanded table below**, showing a strong (though imperfect) correlation between our proposed joint‑likelihood proxy reward $P_{\mathrm{LM}}(v_z, y^\star \mid x, z)$ and veracity accuracy (Hamming similarity).
>
> **Expanded correlation results (Pearson ↑ / Hamming similarity ↑)**
>
> | Model       | ProntoQA        | CommonsenseQA    |
> |-------------|------------------|------------------|
> | Qwen 4B     | 0.56 / **0.91** | 0.72 / **0.94** |
> | Qwen 8B     | 0.74 / **0.95** | 0.78 / **0.93** |
> | LLaMA 3B    | 0.72 / **0.95** | 0.67 / **0.84** |
> | LLaMA 8B    | 0.70 / **0.96** | 0.79 / **0.90** |
>
> For AVI, as an amortized‑inference method, performance is expected to be bounded by that of its supervisory signal (VS). If AVI poorly approximates the target distribution induced by VS, downstream performance will degrade accordingly.
>
> Empirically, Tables 1 and 2 show that (i) VS yields highly accurate veracity labels, and (ii) AVI closely matches VS, generalizing to contexts not seen during training. Figure 4 indicates a favorable accuracy–runtime trade‑off when amortizing VS into AVI, with zero‑shot AVI performance comparable to 100‑iteration VS.

---

> ### Author Response · Authors · 2025-11-21
>
> ### **Q3: How do you think your proposed method will perform on larger or smaller LLMs? Your experiment only involves LLMs of approximately 4B or 8B.**
>
> This is a great question. We would expect models of different sizes to exhibit different verification capabilities, which in turn impacts both VS and in‑context learning baselines (Many2Many). Below, we provide new experimental results on smaller (1.7B) and larger (14B) Qwen models:
>
>
> **GSM8K (1000 samples)**
>
> |  | Qwen 1.7B  |Qwen 14B  |
> | -------- | -------- |-------- |
> | Many2Many   | 0.512       | 0.675   |
> | VS     | **0.657**    | **0.833**      |
>
> **ProntoQA (1000 samples)**
>
> |  | Qwen 1.7B  |Qwen 14B  |
> | -------- | -------- |-------- |
> | Many2Many   | 0.529       |  0.714  |
> | VS     | **0.713**     | **0.928**      |
>
> **CommonsenseQA (1000 samples)**
>
> |  | Qwen 1.7B |Qwen 14B |
> | -------- | -------- |-------- |
> | Many2Many   | 0.488        |  0.524  |
> | VS     | **0.832**     | **0.980**      |
>
> The accuracy of veracity inference improves for both VS and baselines as model capacity increases, with VS continuing to show stronger performance in all cases.

---

### Author Response · Authors · 2025-11-22
**Summary of Revisions**

Based on reviewer feedback, we have updated the manuscript as follows (edits appear in purple):
1. Related work (Section 3): Expanded to include Process Reward Models (PRMs) and prior work on the robustness of self-correction.
2. Conclusion: Added remarks outlining possible extensions that incorporate our method into training-time self-improvement approaches.
3. Stepwise error detection in LM-generated reasoning traces: Added new experiments evaluating Veracity Search (VS) in practical settings where reasoning errors arise naturally during CoT generation (Appendix C.1.2).
4. Self-correction results: Consolidated the ProntoQA and CommonsenseQA self-correction experiments using naturally generated CoTs (Appendix C.2).
5. Correlation analysis: Expanded the Pearson correlation experiments to include more models and datasets (Table 10).
6. Inference-time cost: Added a more detailed analysis of the computational cost of VS (Appendix C.6).
7. Sensitivity to model size: Added VS results on Qwen 1.7B and Qwen 14B to better assess sensitivity to model capacity (Table 11).
8. Appendix reorganization: Reordered appendices for clarity (former C.2→C.1.2; C.3→C.2; C.4→C.3; C.5→C.4; C.6→C.5), added a new Appendix C.6 for the inference-time analysis, and a new Appendix C.7 for VS results on smaller and larger models.

We thank the reviewers for their thoughtful feedback.

---

### Meta-Review · Area_Chair_gft2 · 2026-01-08

**Summary:**

The paper proposes a probabilistic framework where the correctness of reasoning steps in a Chain-of-Thought (CoT) is treated as a latent variable. To identify errors, the authors introduce Veracity Search (VS), a search algorithm (based on Metropolis-Hastings) that infers these latent variables by maximizing the Language Model's joint likelihood of the veracity assignments and the final answer.  They further propose Amortized Veracity Inference (AVI), distilling the search results into a model for faster inference.

**Reviewer Concerns:**

**Addressed:**

**Synthetic vs. Natural Errors:** A major concern shared by reviewers knXB, Qqpf, and xAHw was the reliance on artificially corrupted reasoning chains rather than naturally occurring errors. The authors addressed this comprehensively by running new experiments on ProntoQA and CommonsenseQA using natural CoTs. The results showed VS significantly outperforms the baseline (e.g., 0.92 vs 0.74 Hamming similarity on ProntoQA).


**Computational Cost:** Reviewers knXB, Qqpf, and zvRp worried about the inference time and cost of the search method. The authors clarified that they use KV-caching to reduce the overhead of evaluating proposals and provided wall-clock times (e.g., ~7-13s for 100 iterations). They also ensured fair comparisons by matching the sampling budget of baselines.


**Model Scaling:** Reviewer knXB asked about performance on different model sizes. The authors provided new results for Qwen 1.7B and 14B, showing the method scales well and performance generally improves with model size.


**Comparison to PRMs:** Reviewer zvRp asked why not use Process Reward Models (PRMs). The authors rightly pointed out that standard PRMs require expensive step-level supervision, whereas their method is unsupervised regarding step labels.




**Outstanding:**

**Imperfect Correlation:** Reviewer Qqpf noted that the proxy reward (joint likelihood) correlates with veracity but not perfectly. While the authors showed correlations are generally strong (>0.7 for larger models), this remains an inherent limitation of using the LM as its own judge without external tools.

**Reviewer Scores:**

**Reviewer knXB (Current: 4):** **Score: 6.** This reviewer's main negatives were the lack of natural error experiments, lack of time analysis, and model size questions. The authors provided new data for *all* three points. I believe this would flip their vote to Accept.


**Reviewer Qqpf (Current: 6):** **Score: 6.** Their primary weakness was the "artificially corrupted chains". The new experiments on natural errors  directly resolve this.


**Reviewer zvRp (Current: 6):** **Score: 6.** They might raise their score, but they had a strong stance on the cost (200 passes). However, the rebuttal regarding matched budgets and the distinction from supervised PRMs  defends the paper's position well.


**Reviewer xAHw (Current: 6):** **Score: 6.** Their main weakness was the "synthetic errors" issue , which was fully addressed in the rebuttal.

---

### Decision · Program_Chairs · 2026-01-26

Accept (Poster)